# UNDERSTANDING AND MITIGATING ACCURACY DISPARITY IN REGRESSION

## ABSTRACT

With the widespread deployment of large-scale prediction systems in high-stakes domains, *e.g.*, face recognition, criminal justice, etc., disparity on prediction accuracy between different demographic subgroups has called for fundamental understanding on the source of such disparity and algorithmic intervention to mitigate it. In this paper, we study the accuracy disparity problem in regression. To begin with, we first propose an *error decomposition theorem*, which decomposes the accuracy disparity into the distance between label populations and the distance between conditional representations, to help explain why such accuracy disparity appears in practice. Motivated by this error decomposition and the general idea of distribution alignment with statistical distances, we then propose an algorithm to reduce this disparity, and analyze its game-theoretic optima of the proposed objective function. We conduct experiments on four real-world datasets. The experimental results suggest that our proposed algorithms can effectively mitigate accuracy disparity while maintaining the predictive power of the regression models.

## 1 INTRODUCTION

Recent progress in machine learning has led to its widespread use in many high-stakes domains, such as criminal justice, healthcare, student loan approval, and hiring. Meanwhile, it has also been widely observed that accuracy disparity could occur inadvertently under various scenarios in practice (Barocas & Selbst, 2016). For example, errors are inclined to occur for individuals of certain underrepresented demographic groups (Kim, 2016). In other cases, Buolamwini & Gebru (2018) showed that notable accuracy disparity gaps exist across different racial and gender demographic subgroups on several real-world image classification systems. Moreover, Bagdasaryan et al. (2019) found out that a differentially private model even enlarges such accuracy disparity gaps. Such accuracy disparity gaps across demographic subgroups not only raise concerns in high-stake applications but also can be utilized by malicious parties causing information leakage (Yaghini et al., 2019).

Despite the ample needs of accuracy parity, most prior work limits its scope to studying the problem in binary classification settings (Hardt et al., 2016; Zafar et al., 2017b; Zhao et al., 2019; Jiang et al., 2019). In a seminal work, Chen et al. (2018) analyzed the impact of data collection on accuracy disparity in general learning models. They provided a descriptive analysis of such parity gaps and advocated for collecting more training examples and introducing more predictive variables. While such a suggestion is feasible in applications where data collection and labeling is cheap, it is not applicable in domains where it is time-consuming, expensive, or even infeasible to collect more data, *e.g.*, in autonomous driving, education, etc.

**Our Contributions** In this paper, we provide a prescriptive analysis of accuracy disparity and aim at providing algorithmic interventions to reduce the disparity gap between different demographic subgroups in the regression setting. To start with, we first formally characterize why accuracy disparity appears in regression problems by depicting the feasible region of the underlying group-wise errors. We also provide a lower bound on the joint error and a complementary upper bound on the error gap across groups. Based on these results, we illustrate why regression models aiming to minimize the global loss will inevitably lead to accuracy disparity if the input distributions or decision functions differ across groups (see Figure 1a).

We further propose an error decomposition theorem that decomposes the accuracy disparity into the distance between the label populations and the distance between conditional representations. To mitigate such disparities, we propose two algorithms to reduce accuracy disparity via joint distribution alignment with total variation distance and Wasserstein distance, respectively. Furthermore, we

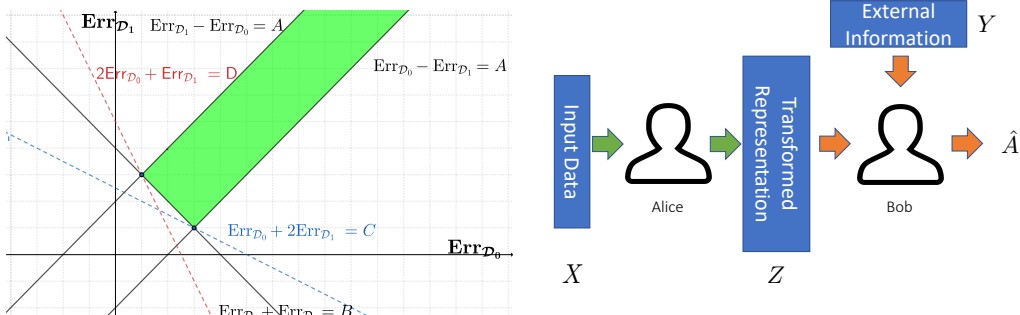

(a) Geometric interpretation of accuracy disparity. (b) Game-theoretic illustration of our algorithms.

Figure 1: The left figure illustrates how accuracy disparity arises by minimizing the global squared loss. The right figure gives a schematic illustration of the proposed algorithmic framework.

analyze the game-theoretic optima of the objective function and illustrate the principle of our algorithms from a game-theoretic perspective (see Figure 1b). To corroborate the effectiveness of our proposed algorithms in reducing accuracy disparity, we conduct experiments on four real-world datasets. Experimental results suggest that our proposed algorithms help to mitigate accuracy disparity while maintaining the predictive power of the regression models. We believe our theoretical results contribute to the understanding of why accuracy disparity occurs in machine learning models, and the proposed algorithms provides an alternative for intervention in real-world scenarios where accuracy parity is desired but collecting more data/features is time-consuming or infeasible.

## 2 PRELIMINARIES

**Notation** We use $\mathcal{X} \subseteq \mathbb{R}^d$ and $\mathcal{Y} \subseteq \mathbb{R}$ to denote the input and output space. We use $X$ and $Y$ to denote random variables which take values in $\mathcal{X}$ and $\mathcal{Y}$, respectively. Lower case letters $\mathbf{x}$ and $y$ denote the instantiation of $X$ and $Y$. We use $H(X)$ to denote the Shannon entropy of random variable $X$, $H(X \mid Y)$ to denote the conditional entropy of $X$ given $Y$, and $I(X; Y)$ to denote the mutual information between $X$ and $Y$. To simplify the presentation, we use $A \in \{0, 1\}$ as the sensitive attribute, *e.g.*, gender, race, etc. Let $\mathcal{H}$ be the hypothesis class of regression models. In other words, for $h \in \mathcal{H}$, $h : \mathcal{X} \rightarrow \mathcal{Y}$ is a predictor. Note that even if the predictor does not explicitly take the sensitive attribute $A$ as an input variable, the prediction can still be biased due to the correlations with other input variables. In this work we study the stochastic setting where there is a joint distribution $\mathcal{D}$ over $X, Y$ and $A$ from which the data are sampled. For $a \in \{0, 1\}$ and $y \in \mathbb{R}$, we use $\mathcal{D}_a$ to denote the conditional distribution of $\mathcal{D}$ given $A = a$ and $\mathcal{D}^y$ to denote the conditional distribution of $\mathcal{D}$ given $Y = y$. For an event $E$, $\mathcal{D}(E)$ denotes the probability of $E$ under $\mathcal{D}$. Given a feature transformation function $g : \mathcal{X} \rightarrow \mathcal{Z}$ that maps instances from the input space $\mathcal{X}$ to feature space $\mathcal{Z}$, we define $g_\sharp \mathcal{D} := \mathcal{D} \circ g^{-1}$ to be the induced (pushforward) distribution of $\mathcal{D}$ under $g$, *i.e.*, for any event $E' \subseteq \mathcal{Z}$, $g_\sharp \mathcal{D}(E') := \mathcal{D}(\{x \in \mathcal{X} \mid g(x) \in E'\})$. We use $(\cdot)_+$ to indicate the value of a variable remains unchanged if it is positive or otherwise 0, *i.e.*, $(Y)_+$ equals to $Y$ if the value of $Y$ is positive or otherwise 0.

Given a joint distribution $\mathcal{D}$, the error of a predictor $h$ under $\mathcal{D}$ is defined as $\mathrm{Err}_\mathcal{D}(h) := \mathbb{E}_\mathcal{D}[(Y - h(X))^2]$. To make the notation more compact, we may drop the subscript $\mathcal{D}$ when it is clear from the context. Furthermore, we also use $\mathrm{MSE}_\mathcal{D}(\widehat{Y}, Y)$ to denote the mean squared loss between the predicted variable $\widehat{Y} = h(X)$ and the true label $Y$ over the joint distribution $\mathcal{D}$. Similarly, we also use $\mathrm{CE}_\mathcal{D}(A \parallel \widehat{A})$ denote the cross-entropy loss between the predicted variable $\widehat{A}$ and the true label $A$ over the joint distribution $\mathcal{D}$. Throughout the paper, we make the following standard assumption in regression problems:

**Assumption 2.1.** There exists $M > 0$, such that for any hypothesis $\mathcal{H} \ni h : \mathcal{X} \rightarrow \mathcal{Y}$, $\|h\|_\infty \leq M$ and $|Y| \leq M$.

**Problem Setup** We study the fair regression problem: the goal is to learn a regressor that is fair in the sense that the errors of the regressor are approximately equal across the groups given by the sensitive attribute $A$. We assume that the sensitive attribute $A$ is only available to the learner during

the training phase and is not visible during the inference phase. We would like to point out that there are many other different and important definitions of fairness (Narayanan, 2018) even in the sub-category of group fairness, and our discussion is by no means comprehensive. For example, two frequently used definitions of fairness in the literature are the so-called statistical parity (Dwork et al., 2012) and equalized odds (Hardt et al., 2016). Nevertheless, throughout this paper we mainly focus accuracy parity as our fairness notion, due to the fact that machine learning systems have been shown to exhibit substantial accuracy disparities between different demographic subgroups (Barocas & Selbst, 2016; Kim, 2016; Buolamwini & Gebru, 2018). This observation has already brought huge public attention (*e.g.*, see New York Times, The Verge, and Insurance Journal) and calls for machine learning systems that (at least approximately) satisfy accuracy parity. Formally, accuracy parity is defined as follows:

**Definition 2.1** (Accuracy Parity). Given a joint distribution $\mathcal{D}$, a predictor $h$ satisfies accuracy parity if $\mathrm{Err}_{\mathcal{D}_0}(h) = \mathrm{Err}_{\mathcal{D}_1}(h)$.

The violation of accuracy parity is also known as *disparate mistreatment* (Zafar et al., 2017a). In practice the exact equality of on accuracy between two groups is often hard to ensure, so we define *error gap* to measure how well the model satisfies accuracy parity:

**Definition 2.2** (Error Gap). Given a joint distribution $\mathcal{D}$, the error gap of a hypothesis $h$ is $\Delta_{\mathrm{Err}}(h) := |\mathrm{Err}_{\mathcal{D}_0}(h) - \mathrm{Err}_{\mathcal{D}_1}(h)|$.

By definition, if a model satisfies accuracy parity, $\Delta_{\mathrm{Err}}(h)$ will be zero. Next we introduce two distance metrics that will be used in our theoretical analysis and algorithm design:

- Total variation distance: it measures the largest possible difference between the probabilities that the two probability distributions can assign to the same event $E$. We use $d_{\mathrm{TV}}(\mathcal{P}, \mathcal{Q})$ to denote the total variation:

$$d_{\mathrm{TV}}(\mathcal{P}, \mathcal{Q}) := \sup_E |\mathcal{P}(E) - \mathcal{Q}(E)|.$$

- Wasserstein distance: the Wasserstein distance between two probability distributions is

$$W_1(\mathcal{P}, \mathcal{Q}) = \sup_{f \in \{f : \|f\|_L \leq 1\}} \left| \int_\Omega f d\mathcal{P} - \int_\Omega f d\mathcal{Q} \right|,$$

where $\|f\|_L$ is the Lipschitz semi-norm of a real-valued function of $f$ and $\Omega$ is the sample space over which two probability distributions $\mathcal{P}$ and $\mathcal{Q}$ are defined. By the Kantorovich-Rubinstein duality theorem (Villani, 2008), we recover the primal form of the Wasserstein distance, defined as

$$W_1(\mathcal{P}, \mathcal{Q}) := \inf_{\gamma \in \Gamma(\mathcal{P}, \mathcal{Q})} \int d(X, Y) \, \mathrm{d}\gamma(X, Y),$$

where $\Gamma(\mathcal{P}, \mathcal{Q})$ denotes the collection of all couplings of $\mathcal{P}$ and $\mathcal{Q}$, and $X$ and $Y$ denote the random variables with law $\mathcal{P}$ and $\mathcal{Q}$ respectively. Note that we use $L_1$ distance for $d(\cdot, \cdot)$ throughout the paper, but the extensions to other distance, *e.g.*, $L_2$ distance, is straightforward.

## 3    MAIN RESULTS

In this section, we first characterize why accuracy disparity arises in regression models. More specifically, given a hypothesis $h \in \mathcal{H}$, we first describe the feasible region of $\mathrm{Err}_{\mathcal{D}_0}$ and $\mathrm{Err}_{\mathcal{D}_1}$ by proving a lower bound of joint errors and an upper bound of the error gap. Then, we give a geometric interpretation to visualize the feasible region of $\mathrm{Err}_{\mathcal{D}_0}$ and $\mathrm{Err}_{\mathcal{D}_1}$ and illustrate how error gap arises when learning a hypothesis $h$ that minimizes the global squared error. We further analyze the accuracy disparity by decomposing it into the distance between label populations and the distance between conditional representations. Motivated by the decomposition, we propose two algorithms to reduce accuracy disparity, connect the game-theoretic optima of the objective functions in our algorithms with our theorems, and describe the practical implementations of the algorithms. Due to the space limit, we defer all the detailed proofs to the appendix.

### 3.1    BOUNDS ON CONDITIONAL ERRORS AND ACCURACY DISPARITY GAP

When we learn a predictor, the prediction function induces $X \xrightarrow{h} \widehat{Y}$, where $\widehat{Y}$ is the predicted target variable given by hypothesis $h$. Hence for any distribution $\mathcal{D}_0$ ($\mathcal{D}_1$) of $X$, the predictor also induces a

distribution $h_\sharp \mathcal{D}_0$ ($h_\sharp \mathcal{D}_1$) of $\widehat{Y}$. Recall that the Wasserstein distance is metric, hence the following chain of triangle inequalities holds:

$$W_1(\mathcal{D}_0(Y), \mathcal{D}_1(Y)) \leq W_1(\mathcal{D}_0(Y), h_\sharp \mathcal{D}_0) + W_1(h_\sharp \mathcal{D}_0, h_\sharp \mathcal{D}_1) + W_1(h_\sharp \mathcal{D}_1, \mathcal{D}_1(Y))$$

Intuitively, $W_1(\mathcal{D}_0(Y), h_\sharp \mathcal{D}_0)$ and $W_1(h_\sharp \mathcal{D}_1, \mathcal{D}_1(Y))$ measure the distance between the true label distribution and the predicted one on $A = 0/1$ cases, respectively. This distance is related to the prediction error of function $h$ conditioned on $A = a$:

**Lemma 3.1.** Let $\widehat{Y} = h(X) \in \mathbb{R}$, then for $a \in \{0, 1\}$, $W_1(\mathcal{D}_a(Y), h_\sharp \mathcal{D}_a) \leq \sqrt{\text{Err}_{\mathcal{D}_a}(h)}$.

With the above results, we can get the following theorem that characterizes the lower bound of joint error on different groups:

**Theorem 3.1.** Let $\widehat{Y} = h(X) \in \mathbb{R}$, we have $\text{Err}_{\mathcal{D}_0}(h) + \text{Err}_{\mathcal{D}_1}(h) \geq \frac{1}{2}\left[\left(W_1(\mathcal{D}_0(Y), \mathcal{D}_1(Y)) - W_1(h_\sharp \mathcal{D}_0, h_\sharp \mathcal{D}_1)\right)_+\right]^2$.

In Theorem 3.1, we see that if the difference between the label distribution across groups is large, then statistical disparity could potentially lead to a large joint error. Moreover, Theorem 3.1 could be extended to give a lower bound on the joint error incurred by $h$ as well:

**Corollary 3.1.** Let $\widehat{Y} = h(X) \in \mathbb{R}$ and $\alpha = \mathcal{D}(A = 0) \in [0, 1]$, we have $\text{Err}_{\mathcal{D}}(h) \geq \frac{1}{2} \min\{\alpha, 1 - \alpha\} \cdot \left[\left(W_1(\mathcal{D}_0(Y), \mathcal{D}_1(Y)) - W_1(h_\sharp \mathcal{D}_0, h_\sharp \mathcal{D}_1)\right)_+\right]^2$.

Next, we upper bound the error gap to gain more insights on accuracy disparity. For $a \in \{0, 1\}$, define the conditional variance $\text{Var}_{\mathcal{D}_a}[Y|X] = \mathbb{E}_{\mathcal{D}_a}[(Y - \mathbb{E}_{\mathcal{D}_a}[Y|X])^2|X]$ and it shows up as the irreducible error of predicting $Y$ when we only use the knowledge of $X$. We also know that the optimal decision function conditioned on $A = a$ under mean squared error to be $\mathbb{E}_{\mathcal{D}_a}[Y|X]$. The following theorem characterizes the upper bound of the error gap between two groups:

**Theorem 3.2.** For any hypothesis $\mathcal{H} \ni h : \mathcal{X} \to \mathcal{Y}$, if the Assumption 2.1 holds, then:

$$\Delta_{\text{Err}}(h) \leq 8M^2\, d_{\text{TV}}(\mathcal{D}_0(X), \mathcal{D}_1(X)) + |\mathbb{E}_{\mathcal{D}_0}[\text{Var}_{\mathcal{D}_0}[Y|X]] - \mathbb{E}_{\mathcal{D}_1}[\text{Var}_{\mathcal{D}_1}[Y|X]]|$$
$$+ 4M\,\min\{\mathbb{E}_{\mathcal{D}_0}[|\mathbb{E}_{\mathcal{D}_0}(Y|X) - \mathbb{E}_{\mathcal{D}_1}(Y|X)|], \mathbb{E}_{\mathcal{D}_1}[|\mathbb{E}_{\mathcal{D}_0}(Y|X) - \mathbb{E}_{\mathcal{D}_1}(Y|X)|]\}.$$

**Remark** Theorem 3.2 upper bounds the error gap across groups by three terms: the first term corresponds to the distance of input distribution across groups, the second term is the noise (variance) difference, and third term is the discrepancy of optimal decision functions across different groups. In an ideal and fair setting, where both distributions are noiseless, and the optimal decision functions are insensitive to the group membership, then Theorem 3.2 implies a sufficient condition to guarantee accuracy parity is to find group-invariant representation that minimize $d_{\text{TV}}(\mathcal{D}_0(X), \mathcal{D}_1(X))$.

**Geometric Interpretation** By Theorem 3.1 and Theorem 3.2, in Figure 1a, we visually illustrate how accuracy disparity arises given data distribution and the learned hypothesis that aims to minimize the global squared error. In Figure 1a, given the hypothesis class $\mathcal{H}$, we use the line $\text{Err}_{\mathcal{D}_0} + \text{Err}_{\mathcal{D}_1} = B$ to denote the lower bound in Theorem 3.1 and the two lines $|\text{Err}_{\mathcal{D}_0} - \text{Err}_{\mathcal{D}_1}| = A$ to denote the upper bound in Theorem 3.2. These three lines form a feasible region (the green area) of $\text{Err}_{\mathcal{D}_0}$ and $\text{Err}_{\mathcal{D}_1}$ under the hypothesis class $\mathcal{H}$. For any optimal hypothesis $h$ which is solely designed to minimize the overall error, the best the hypothesis $h$ can do is to intersect with one of the two bottom vertices. For example, the hypotheses (the red dotted line and the blue dotted line) trying to minimize overall error intersect with the two vertices of the region to achieve the smallest $\text{Err}_{\mathcal{D}_0}$-intercept ($\text{Err}_{\mathcal{D}_1}$-intercept), due to the imbalance between these two groups. However, since these two vertices are not on the diagonal of the feasible region, there is no guarantee that the hypothesis can satisfy accuracy parity ($\text{Err}_{\mathcal{D}_0} = \text{Err}_{\mathcal{D}_1}$), unless we can shrink the width of green area to zero.

**Conditional Distribution Alignment Reduces Accuracy Parity** In Theorem 3.2, we illustrate how accuracy disparity arises in regression models due to noise, distance between representations, and distance between decision functions. However, it is nearly impossible to collect noiseless data with group-invariant input distribution. Moreover, there is no guarantee that the upper bound will be lower if we learn the group-invariant representation that minimizes $d_{\text{TV}}(\mathcal{D}_0(X), \mathcal{D}_1(X))$ alone, since the learned representation could potentially increase the variance. In this regard, we prove a novel upper bound which is free from the above noise term to motivate aligning conditional distributions to mitigate the error disparity across groups. To do so, we relate the error gap to the label distribution and the predicted distribution condition on $Y = y$:

**Theorem 3.3.** If Assumption 2.1 holds, then for $\forall h \in \mathcal{H}$, let $\widehat{Y} = h(X)$, the following inequality holds:

$$\Delta_{\mathrm{Err}}(h) \leq 8M^2 d_{\mathrm{TV}}(\mathcal{D}_0(Y), \mathcal{D}_1(Y))$$
$$+ 3M \min\{\mathbb{E}_{\mathcal{D}_0}[|\mathbb{E}_{\mathcal{D}_0^y}[\widehat{Y}] - \mathbb{E}_{\mathcal{D}_1^y}[\widehat{Y}]|], \ \mathbb{E}_{\mathcal{D}_1}[|\mathbb{E}_{\mathcal{D}_0^y}[\widehat{Y}] - \mathbb{E}_{\mathcal{D}_1^y}[\widehat{Y}]|]\}.$$

**Remark** We see that the error gap is upper bounded by two terms: the distance between label distributions and the discrepancy between conditional predicted distributions across groups. Note that this is different from the decomposition we have in Theorem 3.2, where the marginal distribution is on $X$ instead of $Y$. Given a dataset, the distance of label distributions is a constant since the label distribution is fixed. For the second term, if we can minimize the discrepancy of the conditional predicted distribution across groups, we then have a model that is free of accuracy disparity when the label distribution is well aligned.

## 3.2 ALGORITHM DESIGN

Inspired by Theorem 3.3, we can mitigate the error gap if we align the group distributions via minimizing the distance of the conditional distributions across groups. However, it is intractable to do so explicitly in regression problems since $Y$ can take infinite values on $\mathbb{R}$. Next we will present two algorithms to approximately solve the problem through adversarial representation learning.

Given a Markov chain $X \xrightarrow{g} Z \xrightarrow{h} \widehat{Y}$, we are interested in learning group-invariant conditional representations so that the discrepancy between the induced conditional distributions $\mathcal{D}_0^Y(Z = g(X))$ and $\mathcal{D}_1^Y(Z = g(X))$ is minimized. In this case, the second term of the upper bound in Theorem 3.3 is minimized. However, it is in general not feasible since $Y$ is a continuous random variable. Instead, we propose to learn the representations of $Z$ to minimize the discrepancy between the joint distributions $\mathcal{D}_0(Z = g(X), Y)$ and $\mathcal{D}_1(Z = g(X), Y)$. Next, we will show the distances between conditional predicted distributions $\mathcal{D}_0^Y(Z = g(X))$ and $\mathcal{D}_1^Y(Z = g(X))$ are minimized when we minimize the joint distributions $\mathcal{D}_0(Z = g(X), Y)$ and $\mathcal{D}_1(Z = g(X), Y)$ in Theorem 3.4 and Theorem 3.5.

To proceed, we first consider using the total variation distance to measure the distance between two distributions. In particular, we can choose to learn a binary discriminator $f : Z \times Y \longrightarrow \widehat{A}$ that achieves minimum binary classification error on discriminating between points sampled from two distributions. In practice, we use the cross-entropy loss as a convex surrogate loss. Formally, we are going to consider the following minimax game between $g$ and $f$:

$$\min_{f \in \mathcal{F}} \max_{g} \quad \mathrm{CE}_{\mathcal{D}}(A \parallel f(g(X), Y)) \tag{1}$$

Next we show that for the above equation, the optimal feature transformation $g$ corresponds to the one that induces invariant conditional feature distributions.

**Theorem 3.4.** Consider the minimax game in (1). The equilibrium $(g^*, f^*)$ of the game is attained when 1). $Z = g^*(X)$ is independent of $A$ conditioned on $Y$; 2). $f^*(Z, Y) = \mathcal{D}(A = 1 \mid Y, Z)$.

Since in the equilibrium of the game $Z$ is independent of $A$ conditioned on $Y$, the optimal $f^*(Z, Y)$ could also be equivalently written as $f^*(Z, Y) = \mathcal{D}(A = 1 \mid Y)$, *i.e.*, the only useful information for the discriminator in the equilibrium is through the external information $Y$. In Theorem 3.4, the minimum cross-entropy loss that the discriminator (the equilibrium of the game) can achieve is $H(A \mid Z, Y)$ (see Proposition A.1 in Appendix A). By the basic property of conditional entropy, we have:

$$\min_{f \in \mathcal{F}} \mathrm{CE}_{\mathcal{D}}(A \parallel f(g(X), Y)) = H(A \mid Z, Y) = H(A \mid Y) - I(A; Z \mid Y).$$

We know that $H(A \mid Y)$ is a constant given the data distribution. The maximization of $g$ in (1) is equivalent to the minimization of $\min_{Z=g(X)} I(A; Z \mid Y)$, and it follows that the optimal strategy for the transformation $g$ is the one that induces conditionally invariant features, *e.g.*, $I(A; Z \mid Y) = 0$. Formally, we arrive at the following minimax problem:

$$\min_{h,g} \max_{f \in \mathcal{F}} \quad \mathrm{MSE}_{\mathcal{D}}(h(g(X)), Y) - \lambda \cdot \mathrm{CE}_{\mathcal{D}}(A \parallel f(g(X), Y)) \tag{2}$$

In the above formulation, the first term corresponds to the minimization of prediction loss of the target task and the second term is the loss incurred by the adversary $f$. As a whole, the minimax

optimization problem expresses a trade-off (controlled by the hyper-parameter $\lambda > 0$) between accuracy and accuracy disparity through the representation learning function $g$.

**Wasserstein Variant** Similarly, if we choose to align joint distributions via minimizing Wasstertein distance, the following theorem holds.

**Theorem 3.5.** Let $g^* := \arg\min_g W_1(\mathcal{D}_0(g(X), Y), \mathcal{D}_1(g(X), Y))$, then $\mathcal{D}_0^Y(Z = g^*(X)) = \mathcal{D}_1^Y(Z = g^*(X))$ almost surely.

One notable advantage of using the Wasserstein distance instead of the TV distance is that, the Wasserstein distance is a continuous functional of both the feature map $g$ as well as the discriminator $f$ (Arjovsky et al., 2017). Furthermore, if both $g$ and $f$ are continuous functions of their corresponding model parameters, (which is the case for models we are going to use in experiments), the objective function will be continuous in both model parameters. This property of the Wasserstein distance makes it more favorable from an optimization perspective. Using the dual formulation, equivalently, we can learn a Lipschitz function $f : Z \times Y \to \mathbb{R}$ as a witness function:

$$\min_{h,g,Z_0 \sim g_\sharp \mathcal{D}_0, Z_1 \sim g_\sharp \mathcal{D}_1} \max_{f : \|f\|_L \leq 1} \text{MSE}_\mathcal{D}(h(g(X)), Y) + \lambda \cdot \big| f(Z_0, Y) - f(Z_1, Y) \big|. \quad (3)$$

**Game-Theoretic Interpretation** To make our algorithms easier to follow, we provide a game-theoretic interpretation of our algorithms in Figure 1b. Consider Alice (encoder) and Bob (discriminator) participate a two-player game: upon receiving a set of inputs $X$, Alice applies a transformation to the inputs to generate the corresponding features $Z$ and then send them to Bob. Besides the features sent by Alice, Bob also has access to the external information $Y$, which corresponds to the corresponding labels for the set of features sent by Alice. Once having both the features $Z$ and the corresponding labels $Y$ from external resources, Bob's goal is to guess the group membership $A$ of each feature sent by Alice, and to maximize his correctness as much as possible. On the other hand, Alice's goal is to compete with Bob, *i.e.*, to find a transformation to confuse Bob as much as she can. Different from the traditional game without external information, here due to the external information $Y$ Bob has access to, Alice cannot hope to fully fool Bob, since Bob can gain some insights about the group membership $A$ of features from the external label information. Nevertheless, Theorem 3.4 and Theorem 3.5 both state that when Bob uses a binary discriminator or a Wasstertein discriminator to learn $A$, the best Alice could do is to to learn a transformation $g$ so that the transformed representation $Z$ is insensitive to the values of A conditioned on any values of $Y$.

## 4 EXPERIMENTS

Inspired by our theoretical results that decompose accuracy disparity into the distance between label populations and the distance between conditional representations, we propose two algorithms to mitigate it. In this section, we conduct experiments to evaluate the effectiveness of our proposed algorithms in reducing the accuracy disparity.

### 4.1 EXPERIMENTAL SETUP

**Datasets** We conduct experiments on four real-world benchmark datasets: the Adult dataset (Dua & Graff, 2017), COMPAS dataset (Dieterich et al., 2016), Law School dataset (Wightman & Ramsey, 1998), and Communities and Crime dataset (Dua & Graff, 2017). All datasets contain binary sensitive attributes (*e.g.*, male/female, white/non-white). We refer readers to Appendix B for detailed descriptions of the datasets and the data pre-processing pipelines.

**Methods** We term the proposed algorithms CENET and WASSERSTEINNET for our two proposed algorithms respectively. For each dataset, we perform controlled experiments by fixing the regression neural network architecture to be the same. We train the regression nets via mean squared loss. Note that although the Adult dataset and COMPAS dataset are for binary classification tasks, we can still take them as regression tasks with two distinctive ordinal values. To the best of our knowledge, no previous study aims to minimize accuracy disparity in regression using representation learning. However, there are other similar fairness notions and mitigation techniques proposed for regression and we add them as our baselines: (1) Bounded group loss (BGL) (Agarwal et al., 2019), which asks for the prediction errors for any groups to remain below a pre-defined level $\epsilon$; (2) Coefficient of determination (CoD) (Komiyama et al., 2018), which asks for the coefficient of determination between the sensitive attributes and the predictions to remain below a pre-defined level $\epsilon$.

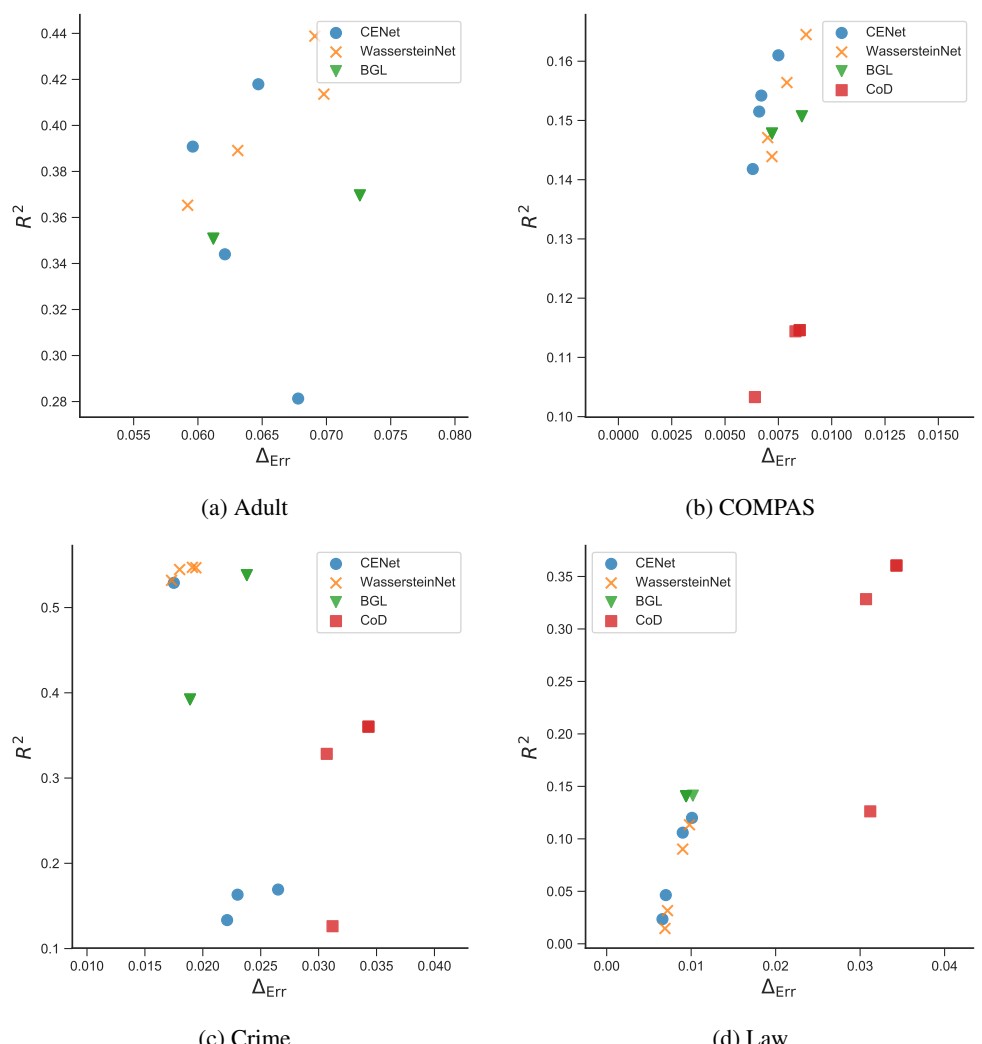

(a) Adult

(b) COMPAS

(c) Crime

(d) Law

Figure 2: Overall results: $R^2$ regression scores and error gaps in different datasets. Our goal is to achieve high $R^2$ scores with small error gap values (*i.e.*, the points located in the upper-left corner).

Among all methods, we vary the trade-off parameter (*i.e.*, $\lambda$ in CENET and WASSERSTEINNET and $\epsilon$ in BGL and COD) and report and the corresponding $R^2$ scores and the error gap values. For each experiment, we average the results for ten random seeds. We refer readers to Appendix B for detailed parameter and hyper-parameter settings in our experiments. We also defer the additional experimental results and analyses on how the trade-off parameters $\lambda$ and $\epsilon$ affects the performance of different algorithms to Appendix C.

### 4.2 RESULTS AND ANALYSES

The overall results are visualized in Figure 2.[1] The following summarizes our observations and analyses: (1) Overall, trade-offs exist between the predictive power of the regressors and accuracy parity: for each method we test, the general trend is that with the decrease of the values of error gaps, the values of $R^2$ also decrease. The exception is CENET in the Adult dataset and Crime dataset since training CENET is unstable when $\lambda$ is large and we will provide more details in Appendix C; (2) Our proposed methods WASSERSTEINNET and CENET are effective in reducing the error gaps while keeping the $R^2$ scores relatively high in the Adult, COMPAS and Crime dataset. In the Law dataset, the error gaps decrease with high utility losses in our proposed methods; (3) Among our

---

[1]COD cannot be implemented on the Adult dataset since the size of the Adult dataset is large and the QCQP optimization algorithm to solve COD needs a quadratic memory usage of the dataset size.

proposed methods, WASSERSTEINNET achieves better accuracy and accuracy disparity trade-offs while CENET suffers significant accuracy loss and may fail to decrease the error gaps in the Adult and Crime dataset. The reason behind it is that the minimax optimization in the training of CENET could lead to an unstable training process under the presence of a noisy approximation to the optimal discriminator (Arjovsky & Bottou, 2017; Arjovsky et al., 2017); (4) Compared to our proposed methods, BGL and COD can also decrease error gaps to a certain extent. This is because: (i) BGL aims to keep errors remaining relatively low in each group, which helps to reduce accuracy disparity; (ii) CoD aims to reduce the correlation between the sensitive attributes and the predictions (or the inputs) in the feature space, which might somehow reduce the dependency between the distributions of these two variables. In comparison, our proposed methods do better in mitigating the error gaps.

## 5 RELATED WORK

**Algorithmic Fairness** In the literature, two main notions of fairness, i.e., *group fairness* and *individual fairness*, has been widely studied (Dwork et al., 2012; Zemel et al., 2013; Feldman et al., 2015; Hardt et al., 2016; Zafar et al., 2017b; Madras et al., 2019; Khani & Liang, 2019). In particular, Chen et al. (2018) analyzed the impact of data collection on discrimination (*e.g.*, false positive rate, false negative rate, and zero-one loss) from the perspectives of bias-variance-noise decomposition, and they suggested collecting more training examples and collect additional variable to reduce discrimination. In comparison, our work precisely characterizes the disparate predictive accuracy in terms of the distance between label populations and the distance between conditional representation and propose algorithms to reduce accuracy disparity across groups in regression.

**Fair Regression** A series of work focus on fairness under the regression problems (Calders et al., 2013; Johnson et al., 2016; Berk et al., 2018; Komiyama et al., 2018; Chzhen et al., 2020; Bigot, 2020; Zink & Rose, 2020; Mary et al., 2019; Narasimhan et al., 2020). To the best of our knowledge, no previous study aimed to minimize accuracy disparity in regression from representation learning. However, there are other similar fairness notions proposed for regression: Agarwal et al. (2019) proposed fair regression with bounded group loss (*i.e.*, it asks that the prediction error for any protected group remain below some pre-defined level) and used exponentiated-gradient approach to satisfy BGL; Komiyama et al. (2018) aimed to reduce the coefficient of determination between the sensitive attributes between the predictions to some pre-defined level and used off-the-shelf convex optimizer to solve the problem. In contrast, we source out the root of accuracy disparity through the lens of information theory and reducing it via distributional alignment in a minimax game.

**Fair Representation** A line of work focus on building algorithmic fair decision making systems using adversarial techniques to learn fair representations (Edwards & Storkey, 2015; Beutel et al., 2017; Adel et al., 2019; Zhao et al., 2019). The main idea behind is to learn a good representation of the data so that the data owner can maximize the accuracy while removing the information related to the sensitive attribute. Madras et al. (2018) proposed a generalized framework to learn adversarially fair and transferable representations and suggests using the label information in the adversary to learn equalized odds or equal opportunity representations in the classification setting. Apart from adversarial representation, recent work also proposed to use distance metrics, *e.g.*, the maximum mean discrepancy (Louizos et al., 2015) and the Wasserstein distance (Jiang et al., 2019) to remove group-related information. Compared to their work, we propose to align (conditional) distributions across groups to reduce accuracy disparity using minimax optimization and analyze the game-theoretic optima in the minimax game in the regression setting.

## 6 CONCLUSION

In this paper, we theoretically and empirically study accuracy disparity in regression problems. Specifically, we prove an information-theoretic lower bound on the joint error and a complementary upper bound on the error gap across groups to depict the feasible region of group-wise errors. Our theoretical results indicate that accuracy disparity occurs inevitably due to the label distributions differ across groups. To reduce such disparity, we further propose to achieve accuracy parity by learning conditional group-invariant representations using statistical distances. The game-theoretic optima of the objective functions in our proposed methods are achieved when the accuracy disparity is minimized. Our empirical results on four real-world datasets demonstrate that our proposed algorithms help to reduce accuracy disparity effectively. We believe our results take an important step towards better understanding accuracy disparity in machine learning models.

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

APPENDIX

In the appendix, we give the proofs of the theorems and claims in our paper, the experimental details and more experimental results.

## A    MISSING PROOFS

**Lemma 3.1.** Let $\widehat{Y} = h(X) \in \mathbb{R}$, then for $a \in \{0, 1\}$, $W_1(\mathcal{D}_a(Y), h_\sharp \mathcal{D}_a) \leq \sqrt{\mathrm{Err}_{\mathcal{D}_a}(h)}$.

*Proof.* The prediction error conditioned on $a \in \{0, 1\}$ is

$$
\begin{aligned}
\mathrm{Err}_{\mathcal{D}_a}(h) =\ & \mathbb{E}[(Y - h(X))^2 | A = a] \\
\geq\ & \mathbb{E}^2[|Y - h(X)| | A = a] \\
\geq\ & \Big( \inf_{\Gamma(\mathcal{D}_a(Y), \mathcal{D}_a(h(X)))} \mathbb{E}[|Y - h(X)|] \Big)^2 \\
=\ & W_1^2(\mathcal{D}_a(Y), h_\sharp \mathcal{D}_a).
\end{aligned}
$$

Taking square root at both sides then completes the proof.  ∎

**Theorem 3.1.** Let $\widehat{Y} = h(X) \in \mathbb{R}$, we have $\mathrm{Err}_{\mathcal{D}_0}(h) + \mathrm{Err}_{\mathcal{D}_1}(h) \geq \frac{1}{2}\big[\big(W_1(\mathcal{D}_0(Y), \mathcal{D}_1(Y)) - W_1(h_\sharp \mathcal{D}_0, h_\sharp \mathcal{D}_1)\big)_+\big]^2$.

*Proof.* Since $W_1(\cdot, \cdot)$ is a distance metric, the result follows immediately the triangle inequality and Lemma 3.1:

$$
W_1(\mathcal{D}_0(Y), \mathcal{D}_1(Y)) \leq \sqrt{\mathrm{Err}_{\mathcal{D}_0}(h)} + W_1(h_\sharp \mathcal{D}_0, h_\sharp \mathcal{D}_1) + \sqrt{\mathrm{Err}_{\mathcal{D}_1}(h)}.
$$

Rearrange the equation above and by AM-GM inequality, we have

$$
W_1(\mathcal{D}_0(Y), \mathcal{D}_1(Y)) - W_1(h_\sharp \mathcal{D}_0, h_\sharp \mathcal{D}_1) \leq \sqrt{\mathrm{Err}_{\mathcal{D}_0}(h)} + \sqrt{\mathrm{Err}_{\mathcal{D}_1}(h)} \leq \sqrt{2(\mathrm{Err}_{\mathcal{D}_0}(h) + \mathrm{Err}_{\mathcal{D}_1}(h))}.
$$

Taking square at both sides then completes the proof.  ∎

**Corollary 3.1.** Let $\widehat{Y} = h(X) \in \mathbb{R}$ and $\alpha = \mathcal{D}(A = 0) \in [0, 1]$, we have $\mathrm{Err}_{\mathcal{D}}(h) \geq \frac{1}{2} \min\{\alpha, 1 - \alpha\} \cdot \big[\big(W_1(\mathcal{D}_0(Y), \mathcal{D}_1(Y)) - W_1(h_\sharp \mathcal{D}_0, h_\sharp \mathcal{D}_1)\big)_+\big]^2$.

*Proof.* The joint error is

$$
\begin{aligned}
& \mathrm{Err}_{\mathcal{D}}(h) \\
=\ & \alpha\, \mathrm{Err}_{\mathcal{D}_0}(h) + (1 - \alpha)\, \mathrm{Err}_{\mathcal{D}_1}(h) \\
\geq\ & \min\{\alpha, 1 - \alpha\}\big(\mathrm{Err}_{\mathcal{D}_0}(h) + \mathrm{Err}_{\mathcal{D}_1}(h)\big) \\
\geq\ & \frac{1}{2} \min\{\alpha, 1 - \alpha\}[(W_1(\mathcal{D}_0(Y), \mathcal{D}_1(Y)) - W_1(h_\sharp \mathcal{D}_0, h_\sharp \mathcal{D}_1))_+]^2. \quad \text{(Theorem 3.1)}
\end{aligned}
$$

∎

**Lemma A.1.** If Assumption 2.1 holds, then the following inequality holds: $|\mathbb{E}_{\mathcal{D}_0}[(h(X) - \mathbb{E}_{\mathcal{D}_1}[Y|X])^2] - \mathbb{E}_{\mathcal{D}_1}[(h(X) - \mathbb{E}_{\mathcal{D}_1}[Y|X])^2]| \leq 8M^2 d_{\mathrm{TV}}(\mathcal{D}_0(X), \mathcal{D}_1(X))$.

*Proof.* First, we know that $\|h(X) - \mathbb{E}_{\mathcal{D}_a}[Y|X]\|_\infty \leq 2M$, $\forall a \in \{0, 1\}$, since $\|h\|_\infty \leq M$ and $|Y| \leq M$. Now it suffices to bound:

$$
\begin{aligned}
& |\mathbb{E}_{\mathcal{D}_0}[(h(X) - \mathbb{E}_{\mathcal{D}_1}[Y|X])^2] - \mathbb{E}_{\mathcal{D}_1}[(h(X) - \mathbb{E}_{\mathcal{D}_1}[Y|X])^2]| \\
=\ & |\langle (h(X) - \mathbb{E}_{\mathcal{D}_1}[Y|X])^2, d\mathcal{D}_0 - d\mathcal{D}_1 \rangle| \\
\leq\ & \|h(X) - \mathbb{E}_{\mathcal{D}_1}[Y|X]\|_\infty^2 \|d\mathcal{D}_0 - d\mathcal{D}_1\|_1 && \text{(Hölder's inequality)} \\
\leq\ & 4M^2 \|d\mathcal{D}_0 - d\mathcal{D}_1\|_1 && \text{(Assumption 2.1)} \\
=\ & 8M^2 d_{\mathrm{TV}}(\mathcal{D}_0(X), \mathcal{D}_1(X)).
\end{aligned}
$$

Note that the last equation follows the definition of total variation distance.  ∎

**Lemma A.2.** If Assumption 2.1 holds, then the following inequality holds: $|\mathbb{E}_{\mathcal{D}_0}[(h(X) - \mathbb{E}_{\mathcal{D}_0}[Y|X])]^2 - \mathbb{E}_{\mathcal{D}_0}[(h(X) - \mathbb{E}_{\mathcal{D}_1}[Y|X])]^2| \leq 4M \mathbb{E}_{\mathcal{D}_0}[|\mathbb{E}_{\mathcal{D}_0}[Y|X] - \mathbb{E}_{\mathcal{D}_1}[Y|X]|]$.

*Proof.*

$$|\mathbb{E}_{\mathcal{D}_0}[(h(X) - \mathbb{E}_{\mathcal{D}_0}[Y|X])]^2 - \mathbb{E}_{\mathcal{D}_0}[(h(X) - \mathbb{E}_{\mathcal{D}_1}[Y|X])]^2|$$
$$= |\mathbb{E}_{\mathcal{D}_0}[h^2(X) - 2h(X)\mathbb{E}_{\mathcal{D}_0}[Y|X] + \mathbb{E}_{\mathcal{D}_0}^2[Y|X] - h^2(X) + 2h(X)\mathbb{E}_{\mathcal{D}_1}[Y|X] - \mathbb{E}_{\mathcal{D}_1}^2[Y|X]]|$$
$$\leq 2M \mathbb{E}_{\mathcal{D}_0}[|\mathbb{E}_{\mathcal{D}_0}[Y|X] - \mathbb{E}_{\mathcal{D}_1}[Y|X]|] + 2M \mathbb{E}_{\mathcal{D}_0}[|\mathbb{E}_{\mathcal{D}_0}[Y|X] - \mathbb{E}_{\mathcal{D}_1}[Y|X]|] \quad \text{(Assumption 2.1)}$$
$$= 4M \mathbb{E}_{\mathcal{D}_0}[|\mathbb{E}_{\mathcal{D}_0}[Y|X] - \mathbb{E}_{\mathcal{D}_1}[Y|X]|].$$

∎

**Theorem 3.2.** For any hypothesis $\mathcal{H} \ni h : \mathcal{X} \to \mathcal{Y}$, if the Assumption 2.1 holds, then:

$$\Delta_{\text{Err}}(h) \leq 8M^2 d_{\text{TV}}(\mathcal{D}_0(X), \mathcal{D}_1(X)) + |\mathbb{E}_{\mathcal{D}_0}[\text{Var}_{\mathcal{D}_0}[Y|X]] - \mathbb{E}_{\mathcal{D}_1}[\text{Var}_{\mathcal{D}_1}[Y|X]]|$$
$$+ 4M \, \min\{\mathbb{E}_{\mathcal{D}_0}[|\mathbb{E}_{\mathcal{D}_0}(Y|X) - \mathbb{E}_{\mathcal{D}_1}(Y|X)|], \, \mathbb{E}_{\mathcal{D}_1}[|\mathbb{E}_{\mathcal{D}_0}(Y|X) - \mathbb{E}_{\mathcal{D}_1}(Y|X)|]\}.$$

*Proof.* First, we show that for $a \in \{0, 1\}$,

$$\text{Err}_{\mathcal{D}_a}(h)$$
$$= \mathbb{E}_{\mathcal{D}_a}[(h(X) - Y)^2]$$
$$= \mathbb{E}_{\mathcal{D}_a}[(h(X) - \mathbb{E}_{\mathcal{D}_a}[Y|X] + \mathbb{E}_{\mathcal{D}_a}[Y|X] - Y)^2]$$
$$= \mathbb{E}_{\mathcal{D}_a}[(h(X) - \mathbb{E}_{\mathcal{D}_a}[Y|X])^2] + \mathbb{E}_{\mathcal{D}_a}[(Y - \mathbb{E}_{\mathcal{D}_a}[Y|X])^2]$$
$$\quad - 2 \mathbb{E}_{\mathcal{D}_a}[(h(X) - \mathbb{E}_{\mathcal{D}_a}[Y|X])(Y - \mathbb{E}_{\mathcal{D}_a}[Y|X])]$$
$$= \mathbb{E}_{\mathcal{D}_a}[(h(X) - \mathbb{E}_{\mathcal{D}_a}[Y|X])^2] + \mathbb{E}_{\mathcal{D}_a}[(Y - \mathbb{E}_{\mathcal{D}_a}[Y|X])^2].$$

Note that the last equation holds since

$$\mathbb{E}_{\mathcal{D}_a}[(h(X) - \mathbb{E}_{\mathcal{D}_a}[Y|X])(Y - \mathbb{E}_{\mathcal{D}_a}[Y|X])]$$
$$= \mathbb{E}_{\mathcal{D}_a(X)}[\mathbb{E}_{\mathcal{D}_a(Y|X)}[(h(X) - \mathbb{E}_{\mathcal{D}_a}[Y|X])(Y - \mathbb{E}_{\mathcal{D}_a}[Y|X])|X]]$$
$$= \mathbb{E}_{\mathcal{D}_a(X)}[(h(X) - \mathbb{E}_{\mathcal{D}_a}[Y|X])\mathbb{E}_{\mathcal{D}_a(Y|X)}(Y - \mathbb{E}_{\mathcal{D}_a}[Y|X]|X)]$$
$$= \mathbb{E}_{\mathcal{D}_a(X)}[(h(X) - \mathbb{E}_{\mathcal{D}_a}[Y|X])(\mathbb{E}_{\mathcal{D}_a}[Y|X] - \mathbb{E}_{\mathcal{D}_a}[Y|X])]$$
$$= 0.$$

Next we bound the error gap:

$$|\text{Err}_{\mathcal{D}_0}(h) - \text{Err}_{\mathcal{D}_1}(h)|$$
$$= |\mathbb{E}_{\mathcal{D}_0}[(h(X) - \mathbb{E}_{\mathcal{D}_0}[Y|X])^2] - \mathbb{E}_{\mathcal{D}_1}[(h(X) - \mathbb{E}_{\mathcal{D}_1}[Y|X])^2]$$
$$\quad + \mathbb{E}_{\mathcal{D}_0}[(Y - \mathbb{E}_{\mathcal{D}_0}[Y|X])^2] - \mathbb{E}_{\mathcal{D}_1}[(Y - \mathbb{E}_{\mathcal{D}_1}[Y|X])^2]|$$
$$\leq |\mathbb{E}_{\mathcal{D}_0}[(h(X) - \mathbb{E}_{\mathcal{D}_0}[Y|X])^2] - \mathbb{E}_{\mathcal{D}_1}[(h(X) - \mathbb{E}_{\mathcal{D}_1}[Y|X])^2]| \quad \text{(Triangle inequality)}$$
$$\quad + |\mathbb{E}_{\mathcal{D}_0}[\text{Var}_{\mathcal{D}_0}[Y|X]] - \mathbb{E}_{\mathcal{D}_1}[\text{Var}_{\mathcal{D}_1}[Y|X]]|.$$

Now it suffices to bound:

$$|\mathbb{E}_{\mathcal{D}_0}[(h(X) - \mathbb{E}_{\mathcal{D}_0}[Y|X])^2] - \mathbb{E}_{\mathcal{D}_1}[(h(X) - \mathbb{E}_{\mathcal{D}_1}[Y|X])^2]|$$
$$= |\mathbb{E}_{\mathcal{D}_0}[(h(X) - \mathbb{E}_{\mathcal{D}_0}[Y|X])^2] - \mathbb{E}_{\mathcal{D}_0}[(h(X) - \mathbb{E}_{\mathcal{D}_1}[Y|X])^2]$$
$$\quad + \mathbb{E}_{\mathcal{D}_0}[(h(X) - \mathbb{E}_{\mathcal{D}_1}[Y|X])^2] - \mathbb{E}_{\mathcal{D}_1}[(h(X) - \mathbb{E}_{\mathcal{D}_1}[Y|X])^2]|$$
$$\leq |\mathbb{E}_{\mathcal{D}_0}[(h(X) - \mathbb{E}_{\mathcal{D}_0}[Y|X])^2] - \mathbb{E}_{\mathcal{D}_0}[(h(X) - \mathbb{E}_{\mathcal{D}_1}[Y|X])^2]| \quad \text{(Triangle inequality)}$$
$$\quad + |\mathbb{E}_{\mathcal{D}_0}[(h(X) - \mathbb{E}_{\mathcal{D}_1}[Y|X])^2] - \mathbb{E}_{\mathcal{D}_1}[(h(X) - \mathbb{E}_{\mathcal{D}_1}[Y|X])^2]|.$$

Invoke Lemma A.1 and Lemma A.2 to bound the above two terms:

$$|\mathbb{E}_{\mathcal{D}_0}[(h(X) - \mathbb{E}_{\mathcal{D}_0}[Y|X])^2] - \mathbb{E}_{\mathcal{D}_0}[(h(X) - \mathbb{E}_{\mathcal{D}_1}[Y|X])^2]|$$
$$+ |\mathbb{E}_{\mathcal{D}_0}[(h(X) - \mathbb{E}_{\mathcal{D}_1}[Y|X])^2] - \mathbb{E}_{\mathcal{D}_1}[(h(X) - \mathbb{E}_{\mathcal{D}_1}[Y|X])^2]|$$
$$\leq 4M \mathbb{E}_{\mathcal{D}_0}[|\mathbb{E}_{\mathcal{D}_0}[Y|X] - \mathbb{E}_{\mathcal{D}_1}[Y|X]|] + 8M^2 d_{\text{TV}}(\mathcal{D}_0(X), \mathcal{D}_1(X)). \quad \text{(Lemma A.1 \& Lemma A.2)}$$

By symmetry, we also have:

$$|\mathbb{E}_{\mathcal{D}_0}[(h(X) - \mathbb{E}_{\mathcal{D}_0}[Y|X])^2] - \mathbb{E}_{\mathcal{D}_0}[(h(X) - \mathbb{E}_{\mathcal{D}_1}[Y|X])^2]|$$
$$+ |\mathbb{E}_{\mathcal{D}_0}[(h(X) - \mathbb{E}_{\mathcal{D}_1}[Y|X])^2] - \mathbb{E}_{\mathcal{D}_1}[(h(X) - \mathbb{E}_{\mathcal{D}_1}[Y|X])^2]|$$
$$\leq 4M\,\mathbb{E}_{\mathcal{D}_1}[|\mathbb{E}_{\mathcal{D}_0}[Y|X] - \mathbb{E}_{\mathcal{D}_1}[Y|X]|] + 8M^2 d_{\mathrm{TV}}(\mathcal{D}_0(X), \mathcal{D}_1(X)). \quad \text{(Lemma A.1 \& Lemma A.2)}$$

Combining the two inequalities above together, we have:

$$|\mathbb{E}_{\mathcal{D}_0}(h(X) - \mathbb{E}_{\mathcal{D}_0}[Y|X])^2 - \mathbb{E}_{\mathcal{D}_0}(h(X) - \mathbb{E}_{\mathcal{D}_1}[Y|X])^2|$$
$$+ |\mathbb{E}_{\mathcal{D}_0}(h(X) - \mathbb{E}_{\mathcal{D}_1}[Y|X])^2 - \mathbb{E}_{\mathcal{D}_1}(h(X) - \mathbb{E}_{\mathcal{D}_1}[Y|X])^2|$$
$$\leq 8M^2 d_{\mathrm{TV}}(\mathcal{D}_0(X), \mathcal{D}_1(X))$$
$$+ 4M \min\{\mathbb{E}_{\mathcal{D}_0}[|\mathbb{E}_{\mathcal{D}_0}[Y|X] - \mathbb{E}_{\mathcal{D}_1}[Y|X]|], \mathbb{E}_{\mathcal{D}_1}[|\mathbb{E}_{\mathcal{D}_0}[Y|X] - \mathbb{E}_{\mathcal{D}_1}[Y|X]|]\}.$$

Incorporating the two variance terms back to the above inequality then completes the proof. ∎

**Theorem 3.3.** If Assumption 2.1 holds, then for $\forall h \in \mathcal{H}$, let $\widehat{Y} = h(X)$, the following inequality holds:

$$\Delta_{\mathrm{Err}}(h) \leq 8M^2 d_{\mathrm{TV}}(\mathcal{D}_0(Y), \mathcal{D}_1(Y))$$
$$+ 3M \min\{\mathbb{E}_{\mathcal{D}_0}[|\mathbb{E}_{\mathcal{D}_0^y}[\widehat{Y}] - \mathbb{E}_{\mathcal{D}_1^y}[\widehat{Y}]|], \ \mathbb{E}_{\mathcal{D}_1}[|\mathbb{E}_{\mathcal{D}_0^y}[\widehat{Y}] - \mathbb{E}_{\mathcal{D}_1^y}[\widehat{Y}]|]\}.$$

*Proof.* First, we show that for $a \in \{0, 1\}$:

$$\mathrm{Err}_{\mathcal{D}_a}(h) = \mathbb{E}_{\mathcal{D}_a}[(h(X) - Y)^2] = \mathbb{E}_{\mathcal{D}_a}[h^2(X) - 2Yh(X) + Y^2] = \mathbb{E}_{\mathcal{D}_a}[h^2(X) - 2Yh(X)] + \mathbb{E}_{\mathcal{D}_a}[Y^2].$$

Next, we bound the error gap:

$$|\mathrm{Err}_{\mathcal{D}_0}(h) - \mathrm{Err}_{\mathcal{D}_1}(h)|$$
$$= |\mathbb{E}_{\mathcal{D}_0}[h^2(X) - 2Yh(X)] + \mathbb{E}_{\mathcal{D}_0}[Y^2] - \mathbb{E}_{\mathcal{D}_1}[h^2(X) - 2Yh(X)] - \mathbb{E}_{\mathcal{D}_1}[Y^2]|$$
$$\leq |\mathbb{E}_{\mathcal{D}_0}[h^2(X) - 2Yh(X)] - \mathbb{E}_{\mathcal{D}_1}[h^2(X) - 2Yh(X)]| + |\mathbb{E}_{\mathcal{D}_0}[Y^2] - \mathbb{E}_{\mathcal{D}_1}[Y^2]|. \quad \text{(Triangle inequality)}$$

For the second term, we can easily prove that

$$|\mathbb{E}_{\mathcal{D}_0}[Y^2] - \mathbb{E}_{\mathcal{D}_1}[Y^2]| = |\langle Y^2, d\mathcal{D}_0 - d\mathcal{D}_1 \rangle| \leq \|Y\|_\infty^2 \|d\mathcal{D}_0 - d\mathcal{D}_1\|_1 \leq 2M^2 d_{\mathrm{TV}}(\mathcal{D}_0(Y), \mathcal{D}_1(Y)),$$

where the second equation follows Hölder's inequality and the last equation follow the definition of total variation distance. Now it suffices to bound the remaining term:

$$|\mathbb{E}_{\mathcal{D}_0}[h^2(X) - 2Yh(X)] - \mathbb{E}_{\mathcal{D}_1}[h^2(X) - 2Yh(X)]|$$
$$= \left| \int h(\mathbf{x})(h(\mathbf{x}) - 2y)\,d\mu_0(\mathbf{x}, y) - \int h(\mathbf{x})(h(\mathbf{x}) - 2y)\,d\mu_1(\mathbf{x}, y) \right|$$
$$\leq \left| \iint h(\mathbf{x})(h(\mathbf{x}) - 2y)\,d\mu_0(\mathbf{x}|y)d\mu_0(y) - \iint h(\mathbf{x})(h(\mathbf{x}) - 2y)\,d\mu_0(\mathbf{x}|y)d\mu_1(y) \right| \quad \text{(Triangle inequality)}$$
$$+ \left| \iint h(\mathbf{x})(h(\mathbf{x}) - 2y)\,d\mu_1(\mathbf{x}|y)d\mu_1(y) - \iint h(\mathbf{x})(h(\mathbf{x}) - 2y)\,d\mu_0(\mathbf{x}|y)d\mu_1(y) \right|.$$

We upper bound the first term:

$$\left| \iint h(\mathbf{x})(h(\mathbf{x}) - 2y)\,d\mu_0(\mathbf{x}|y)\,d\mu_0(y) - \iint h(\mathbf{x})(h(\mathbf{x}) - 2y)\,d\mu_0(\mathbf{x}|y)\,d\mu_1(y) \right|$$
$$\leq \iint \left| h(\mathbf{x})(h(\mathbf{x}) - 2y)(d\mu_0(y) - d\mu_1(y)) \right| d\mu_0(\mathbf{x}|y)$$
$$\leq \int \left| d\mu_0(y) - d\mu_1(y) \right| \int \left| \sup_{\mathbf{x}} h(\mathbf{x}) \right| |h(\mathbf{x}) - 2y|\,d\mu_0(\mathbf{x}|y)$$
$$\leq M \int \mathbb{E}_{\mathcal{D}_0}[|h(X) - 2Y| | Y = y]\,|d\mu_0(y) - d\mu_1(y)| \quad \text{(Assumption 2.1)}$$
$$\leq 3M^2 \int |d\mu_0(y) - d\mu_1(y)| \quad \text{(Assumption 2.1)}$$
$$\leq 6M^2 d_{\mathrm{TV}}(\mathcal{D}_0(Y), \mathcal{D}_1(Y)).$$

Note that the last equation follows the definition of total variation distance. For the second term, we have:

$$\left| \iint h(\mathbf{x})(h(\mathbf{x}) - 2y) \, \mathrm{d}\mu_1(\mathbf{x}|y) \, \mathrm{d}\mu_1(y) - \iint h(\mathbf{x})(h(\mathbf{x}) - 2y) \, \mathrm{d}\mu_0(\mathbf{x}|y) \, \mathrm{d}\mu_1(y) \right|$$

$$\leq \left| \iint h^2(\mathbf{x})(\mathrm{d}\mu_1(\mathbf{x}|y) - \mathrm{d}\mu_0(\mathbf{x}|y)) \, \mathrm{d}\mu_1(y) \right| + \left| \iint 2y \, h(\mathbf{x})(\mathrm{d}\mu_1(\mathbf{x}|y) - \mathrm{d}\mu_0(\mathbf{x}|y)) \, \mathrm{d}\mu_1(y) \right| \quad \text{(Triangle inequality)}$$

$$\leq 3M \, \mathbb{E}_{\mathcal{D}_1}[|\mathbb{E}_{\mathcal{D}_0^y}[\widehat{Y}] - \mathbb{E}_{\mathcal{D}_1^y}[\widehat{Y}]|]. \quad \text{(Assumption 2.1)}$$

To prove the last equation, we first see that:

$$\left| \iint h^2(\mathbf{x})(\mathrm{d}\mu_1(\mathbf{x}|y) - \mathrm{d}\mu_0(\mathbf{x}|y)) \, \mathrm{d}\mu_1(y) \right|$$

$$\leq \left| \iint \left( \sup_{\mathbf{x}} h(\mathbf{x}) \right) h(\mathbf{x})(\mathrm{d}\mu_1(\mathbf{x}|y) - \mathrm{d}\mu_0(\mathbf{x}|y)) \, \mathrm{d}\mu_1(y) \right|$$

$$\leq M \int \left| \mathbb{E}_{\mathcal{D}_0}[h(X)|Y = y] - \mathbb{E}_{\mathcal{D}_1}[h(X)|Y = y] \right| \mathrm{d}\mu_1(y) \quad \text{(Assumption 2.1)}$$

$$= M \, \mathbb{E}_{\mathcal{D}_1}[|\mathbb{E}_{\mathcal{D}_0^y}[\widehat{Y}] - \mathbb{E}_{\mathcal{D}_1^y}[\widehat{Y}]|].$$

Similarly, we also have:

$$\left| \iint 2y \, h(\mathbf{x})(\mathrm{d}\mu_1(\mathbf{x}|y) - \mathrm{d}\mu_0(\mathbf{x}|y)) \, \mathrm{d}\mu_1(y) \right|$$

$$\leq 2 \left| \iint (\sup y) h(\mathbf{x})(\mathrm{d}\mu_1(\mathbf{x}|y) - \mathrm{d}\mu_0(\mathbf{x}|y)) \, \mathrm{d}\mu_1(y) \right|$$

$$\leq 2M \int \left| \mathbb{E}_{\mathcal{D}_0}[h(X)|Y = y] - \mathbb{E}_{\mathcal{D}_1}[h(X)|Y = y] \right| d\mu_1(y) \quad \text{(Assumption 2.1)}$$

$$= 2M \, \mathbb{E}_{\mathcal{D}_1}[|\mathbb{E}_{\mathcal{D}_0^y}[\widehat{Y}] - \mathbb{E}_{\mathcal{D}_1^y}[\widehat{Y}]|].$$

By symmetry, we can also see that:

$$|\mathbb{E}_{\mathcal{D}_0}[h^2(X) - 2Yh(X)] - \mathbb{E}_{\mathcal{D}_1}[h^2(X) - 2Yh(X)]| \leq 6M^2 d_{\text{TV}}(\mathcal{D}_0(Y), \mathcal{D}_1(Y)) + 3M \, \mathbb{E}_{\mathcal{D}_1}[|\mathbb{E}_{\mathcal{D}_0^y}[\widehat{Y}] - \mathbb{E}_{\mathcal{D}_1^y}[\widehat{Y}]|].$$

Combine the above two equations yielding:

$$|\mathbb{E}_{\mathcal{D}_0}[h^2(X) - 2Yh(X)] - \mathbb{E}_{\mathcal{D}_1}[h^2(X) - 2Yh(X)]|$$

$$\leq 6M^2 d_{\text{TV}}(\mathcal{D}_0(Y), \mathcal{D}_1(Y)) + 3M \min\{\mathbb{E}_{\mathcal{D}_0}[|\mathbb{E}_{\mathcal{D}_0^y}[\widehat{Y}] - \mathbb{E}_{\mathcal{D}_1^y}[\widehat{Y}]|], \mathbb{E}_{\mathcal{D}_1}[|\mathbb{E}_{\mathcal{D}_0^y}[\widehat{Y}] - \mathbb{E}_{\mathcal{D}_1^y}[\widehat{Y}]|]\}.$$

Incorporating the terms back to the upper bound of the error gap then completes the proof. ∎

**Theorem 3.4.** Consider the minimax game in (1). The equilibrium $(g^*, f^*)$ of the game is attained when 1). $Z = g^*(X)$ is independent of $A$ conditioned on $Y$; 2). $f^*(Z, Y) = \mathcal{D}(A = 1 \mid Y, Z)$.

*Proof.* To prove Theorem 3.4, we first give Proposition A.1.

**Proposition A.1.** For any feature map $g : \mathcal{X} \to \mathcal{Z}$, assume that $\mathcal{F}$ contains all the randomized binary classifiers and $\mathcal{F} \ni f : \mathcal{Z} \times \mathcal{Y} \to \mathcal{A}$, then $\min_{f \in \mathcal{F}} \text{CE}_{\mathcal{D}}(A \parallel f(g(X), Y)) = H(A \mid Z, Y)$.

*Proof.* By the definition of cross-entropy loss, we have:

$$\begin{aligned} \text{CE}_{\mathcal{D}}(A \parallel f) &= -\mathbb{E}_{\mathcal{D}}\left[\mathbb{I}(A = 0) \log(1 - f(g(X), Y)) + \mathbb{I}(A = 1) \log(f(g(X), Y))\right] \\ &= -\mathbb{E}_{g_\sharp \mathcal{D}}\left[\mathbb{I}(A = 0) \log(1 - f(Z, Y)) + \mathbb{I}(A = 1) \log(f(Z, Y))\right] \\ &= -\mathbb{E}_{Z,Y} \mathbb{E}_{A|Z,Y}\left[\mathbb{I}(A = 0) \log(1 - f(Z, Y)) + \mathbb{I}(A = 1) \log(f(Z, Y))\right] \\ &= -\mathbb{E}_{Z,Y}\left[\mathcal{D}(A = 0 \mid Z, Y) \log(1 - f(Z, Y)) + \mathcal{D}(A = 1 \mid Z, Y) \log(f(Z, Y))\right] \\ &= \mathbb{E}_{Z,Y}\left[D_{\text{KL}}(\mathcal{D}(A \mid Z, Y) \parallel f(Z, Y))\right] + H(A \mid Z, Y) \\ &\geq H(A \mid Z, Y), \end{aligned}$$

where $D_{\text{KL}}(\cdot \| \cdot)$ denotes the KL divergence between two distributions. From the above inequality, it is also clear that the minimum value of the cross-entropy loss is achieved when $f(Z, Y)$ equals the conditional probability $\mathcal{D}(A = 1 \mid Z, Y)$, i.e., $f^*(Z, Y) = \mathcal{D}(A = 1 \mid Z = g(X), Y)$. ∎

Proposition A.1 states that the minimum cross-entropy loss that the discriminator can achieve is $H(A \mid Z, Y)$ when $f$ is the conditional distribution $\mathcal{D}(A = 1 \mid Z = g(X), Y)$. By the basic property of conditional entropy, we have:

$$\min_{f \in \mathcal{F}} \mathrm{CE}_{\mathcal{D}}(A \parallel f(g(X), Y)) = H(A \mid Z, Y) = H(A \mid Y) - I(A; Z \mid Y).$$

Note that $H(A \mid Y)$ is a constant given the distribution $\mathcal{D}$, so the maximization of $g$ is equivalent to the minimization of $\min_{Z=g(X)} I(A; Z \mid Y)$, and it follows that the optimal strategy for the transformation $g$ is the one that induces conditionally invariant features, e.g., $I(A; Z \mid Y) = 0$. On the other hand, if $g^*$ plays optimally, then the optimal response of the discriminator $f$ is given by

$$f^*(Z, Y) = \mathcal{D}(A = 1 \mid Z = g^*(X), Y) = \mathcal{D}(A = 1 \mid Y).$$

∎

**Theorem 3.5.** Let $g^* := \arg\min_g W_1(\mathcal{D}_0(g(X), Y), \mathcal{D}_1(g(X), Y))$, then $\mathcal{D}_0^Y(Z = g^*(X)) = \mathcal{D}_1^Y(Z = g^*(X))$ almost surely.

*Proof.* By the definition of Wasstertein distance, we have:

$$
\begin{aligned}
W_1(\mathcal{D}_0(Z, Y), \mathcal{D}_1(Z, Y)) &= \inf_{\gamma \in \Gamma(\mathcal{D}_0, \mathcal{D}_1)} \int d((\mathbf{z}_0, y_0), (\mathbf{z}_1, y_1)) \, \mathrm{d}\gamma((\mathbf{z}_0, y_0), (\mathbf{z}_1, y_1)) \\
&= \inf_{\gamma \in \Gamma(\mathcal{D}_0, \mathcal{D}_1)} \iint d((\mathbf{z}_0, y_0), (\mathbf{z}_1, y_1)) \, \mathrm{d}\gamma(\mathbf{z}_0, \mathbf{z}_1 \mid y_0, y_1) \, \mathrm{d}\gamma(y_0, y_1) \\
&= \inf_{\gamma \in \Gamma(\mathcal{D}_0, \mathcal{D}_1)} \iint \|\mathbf{z}_0 - \mathbf{z}_1\|_1 + |y_0 - y_1| \, \mathrm{d}\gamma(\mathbf{z}_0, \mathbf{z}_1 \mid y_0, y_1) \, \mathrm{d}\gamma(y_0, y_1) \\
&\geq \inf_{\gamma \in \Gamma(\mathcal{D}_0, \mathcal{D}_1)} \iint |y_0 - y_1| \, \mathrm{d}\gamma(y_0, y_1) \, \mathrm{d}\gamma(\mathbf{z}_0, \mathbf{z}_1 \mid y_0, y_1) \\
&= \inf_{\gamma \in \Gamma(\mathcal{D}_0(Y), \mathcal{D}_1(Y))} \int |y_0 - y_1| \, \mathrm{d}\gamma(y_0, y_1) \\
&= W_1(\mathcal{D}_0(Y), \mathcal{D}_1(Y)).
\end{aligned}
$$

To finish the proof, next we prove the lower bound is achieved when $\mathcal{D}_0^Y(Z = g^*(X)) = \mathcal{D}_1^Y(Z = g^*(X))$: it is easy to see $W_1(\mathcal{D}_0^Y(Z), \mathcal{D}_0^Y(Z)) = \int \|\mathbf{z}_0 - \mathbf{z}_1\|_1 \, \mathrm{d}\gamma(\mathbf{z}_0, \mathbf{z}_1 \mid y_0, y_1) = 0$ when the conditional distributions are equal. In this case, when the Wasserstein distance is minimized, then $Z$ is conditionally independent of $A$ given $Y$. ∎

## B EXPERIMENTAL DETAILS

**Adult** The Adult dataset contains 48,842 examples for income prediction. The task is to predict whether the annual income of an individual is greater or less than 50K/year based on the attributes of the individual, such as education level, age, occupation, etc. In our experiment, we use gender (binary) as the sensitive attribute. The target variable (income) is an ordinal binary variable: 0 if $<$ 50K/year otherwise 1. After data pre-processing, the dataset contains 30,162/15,060 training/test instances where the input dimension of each instance is 113. We show the data distributions for different demographic subgroups in Table 1.

To preprocess the dataset, we first filter out the data records that contain the missing values. We then remove the sensitive attribute from the input features and normalize the input features with its means and standard deviations. Note that we use one-hot encoding for the categorical attributes.

For our proposed methods, we use a three-layer neural network with ReLU as the activation function of the hidden layers and the sigmoid function as the output function for the prediction task (we take the first two layers as the feature mapping). The number of neurons in the hidden layers is 60. We train the neural networks with the ADADELTA algorithm with the learning rate 0.1 and a batch size of 512. The models are trained in 50 epochs. For the adversary networks in CENET and WASSERSTEINNET, we use a two-layer neural network with ReLU as the activation function. The number of neurons in the hidden layers of the adversary networks is 60. The adversary network in CENET also use sigmoid function as the output function. The weight clipping norm in the adversary

network of WASSERSTEINNET is 0.005. We use the gradient reversal layer (Ganin et al., 2016) to implement the gradient descent ascent (GDA) algorithm for optimization of the minimax problem since it makes the training process more stable (Daskalakis & Panageas, 2018). For the rest of the datasets we used in our experiments, we also use gradient reversal layer to implement our algorithms.

We use the Fairlearn toolkit (Bird et al., 2020) to implement BGL: we use the exponentiated-gradient algorithm with the default setting as the mitigator and vary the upper bound $\epsilon \in \{0.07, 0.1, 0.2, 0.5\}$ of the bounded group loss constraint. For each value of $\epsilon$, we run ten random seeds and compute the means and standard deviations.

**COMPAS** The COMPAS dataset 6,172 instances to predict whether a criminal defendant will recidivate within two years or not. It contains attribute such as age, race, etc. In our experiment, we use race (white or non-white) as the sensitive attribute and recidivism as the target variable. We split the dataset into training and test set with the ratio 7/3. We show the data distributions for different demographic subgroups in Table 2.

For all methods, we use a two-layer neural network with ReLU as the activation function of the hidden layers and the sigmoid function as the output function for the prediction task (we take the first layer as the feature mapping). The number of neurons in the hidden layers is 60. We train the neural networks with the ADADELTA algorithm with the learning rate 1.0 and a batch size of 512. The models are trained in 50 epochs. For the adversary networks in CENET and WASSERSTEINNET, we use a two-layer neural network with ReLU as the activation function. The number of neurons in the hidden layers of the adversary networks is 10. The adversary network in CENET also use sigmoid function as the output function. The weight clipping norm in the adversary network of WASSERSTEINNET is 0.05.

We use the Fairlearn toolkit to implement BGL: we use the exponentiated-gradient algorithm with the default setting as the mitigator and vary the upper bound $\epsilon \in \{0.1, 0.2, 0.3, 0.5\}$ of the bounded group loss constraint. For each value of $\epsilon$, we run ten random seeds and compute the means and standard deviations.

As for COD, we follow the source implementation.[2] We use the same hyper-parameter settings as (Komiyama et al., 2018): We use the kernelized optimization with the random Fourier features and the RBF kernel (we vary hyper-parameter of the RBF kernel $\gamma \in \{0.1, 1.0, 10, 100\}$) and report the best results with minimal MSE loss for each time we change the fairness budget $\epsilon$. We also vary $\epsilon \in \{0.01, 0.1, 0.5, 1.0\}$ and run ten random seeds and compute the means and standard deviations.

Table 1: Data distribution of $Y$ and $A$ in Adult dataset.

|  | $Y = 0$ | $Y = 1$ |
|---|---|---|
| $A = 0$ | 20988 | 9539 |
| $A = 1$ | 13026 | 1669 |

Table 2: Data distribution of $Y$ and $A$ in COMPAS dataset.

|  | $Y = 0$ | $Y = 1$ |
|---|---|---|
| $A = 0$ | 1849 | 1148 |
| $A = 1$ | 1514 | 1661 |

**Communities and Crime** The Communities and Crime dataset contains 1,994 examples of socio-economic, law enforcement, and crime data about communities in the United States. The task is to predict the number of violent crimes per 100K population. All attributes in the dataset have been curated and normalized to $[0, 1]$. In our experiment, we use race (binary) as the sensitive attribute: 1 if the population percentage of the white is greater or equal to 80% otherwise 0. After data pre-processing, the dataset contains 1,595/399 training/test instances where the input dimension of each instance is 96. We visualize the data distributions for different demographic subgroups in Figure 3b.

To preprocess the dataset, we first remove the non-predictive attributes and sensitive attributes from the input features. Note that all features in the dataset have already been normalized in $[0, 1]$ so that we do not perform additional normalization to the features. We then replace the missing values with the mean values of the corresponding attributes.

For all methods, we use a two-layer neural network with ReLU as the activation function of the hidden layers and the sigmoid function as the output function for the prediction task (we take the first

---

[2]https://github.com/jkomiyama/fairregresion

layer as the feature mapping). The number of neurons in the hidden layers is 50. We train the neural networks with the ADADELTA algorithm with the learning rate 0.1 and a batch size of 256. The models are trained in 100 epochs. For the adversary networks in CENET and WASSERSTEINNET, we use a two-layer neural network with ReLU as the activation function. The number of neurons in the hidden layers of the adversary networks is 100. The adversary network in CENET also use sigmoid function as the output function. The weight clipping norm in the adversary network of WASSERSTEINNET is 0.002.

We use the Fairlearn toolkit to implement BGL: we use the exponentiated-gradient algorithm with the default setting as the mitigator and vary the upper bound $\epsilon \in \{0.01, 0.02, 0.03, 0.05\}$ of the bounded group loss constraint. For each value of $\epsilon$, we run ten random seeds and compute the means and standard deviations.

As for CoD, we follow the same hyper-parameter settings as (Komiyama et al., 2018): We use the kernelized optimization with the random Fourier features and the RBF kernel (we vary hyper-parameter of the RBF kernel $\gamma \in \{0.1, 1.0, 10, 100\}$) and report the best results with minimal MSE loss for each time we change the fairness budget $\epsilon$. The hyper-parameter settings follow from (Komiyama et al., 2018). We also vary $\epsilon \in \{0.01, 0.1, 0.5, 1.0\}$ and run ten random seeds and compute the means and standard deviations.

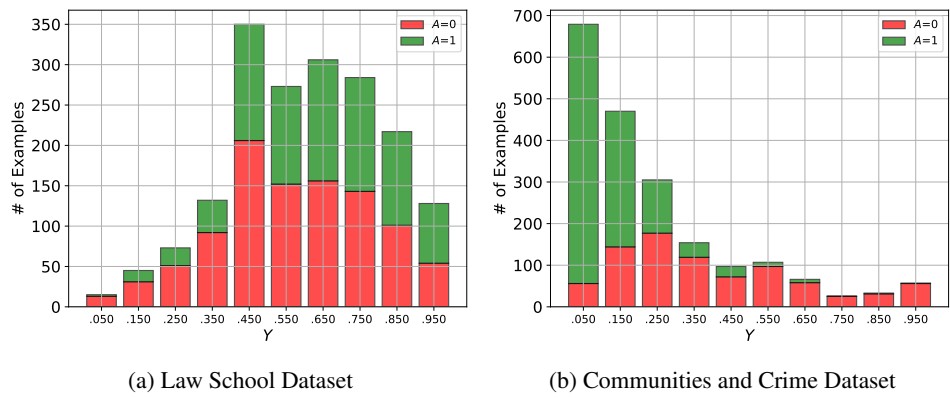

(a) Law School Dataset        (b) Communities and Crime Dataset

Figure 3: Data distributions for different demographic subgroups in two datasets.

**Law School** The Law School dataset contains 1,823 records for law students who took the bar passage study for Law School Admission[3]. The features in the dataset include variables such as undergraduate GPA, LSAT score, full-time status, family income, gender, etc. In our experiment, we use gender as the sensitive attribute and undergraduate GPA as the target variable. We split the dataset into training and test set with the ratio 8/2. We show the data distributions for different demographic subgroups in Figure 3a.

For all methods, we use a two-layer neural network with ReLU as the activation function of the hidden layers and the sigmoid function as the output function for the prediction task (we take the first layer as the feature mapping). The number of neurons in the hidden layers is 10. We train the neural networks with the ADADELTA algorithm with the learning rate 0.1 and a batch size of 256. The models are trained in 100 epochs. For the adversary networks in CENET and WASSERSTEINNET, we use a two-layer neural network with ReLU as the activation function. The number of neurons in the hidden layers of the adversary networks is 10. The adversary network in CENET also use sigmoid function as the output function. The weight clipping norm in the adversary network of WASSERSTEINNET is 0.2.

We use the Fairlearn toolkit to implement BGL: we use the exponentiated-gradient algorithm with the default setting as the mitigator and vary the upper bound $\epsilon \in \{0.01, 0.02, 0.03, 0.05\}$ of the bounded group loss constraint. For each value of $\epsilon$, we run ten random seeds and compute the means and standard deviations.

---

[3]We use the edited public version of the dataset which can be download here: `https://github.com/algowatchpenn/GerryFair/blob/master/dataset/lawschool.csv`

As for CoD, we follow the same hyper-parameter settings as (Komiyama et al., 2018): We use the kernelized optimization with the random Fourier features and the RBF kernel (we vary hyper-parameter of the RBF kernel $\gamma \in \{0.1, 1.0, 10, 100\}$) and report the best results with minimal MSE loss for each time we change the fairness budget $\epsilon$. The hyper-parameter settings follow from (Komiyama et al., 2018). We also vary $\epsilon \in \{0.01, 0.1, 0.5, 1.0\}$ and run ten random seeds and compute the means and standard deviations.

# C  ADDITIONAL EXPERIMENTAL RESULTS AND ANALYSES

In this section, we provide additional experimental results and analyses.

## C.1  IMPACT OF FAIRNESS TRADE-OFF PARAMETERS

We present additional experimental results and analyses to gain more insights into how the fairness trade-off parameters (*e.g.*, $\lambda$ and $\epsilon$) affect the performance of the model predictive performance and accuracy disparity in each methods.

Table 3: $R^2$ regression scores and error gaps when $\lambda$ changes in CENET and WASSERSTEINNET.

**Adult**

| | | $\lambda$ | 0.0 | 1.0 | 10 | 50 | 100 |
|---|---|---|---|---|---|---|---|
| Adult | $R^2$ | CENET | 0.4419±0.0024 | 0.4179±0.0019 | 0.3908±0.0136 | 0.3440±0.0210 | 0.2813±0.0215 |
| | | WASSERSTEINNET | 0.4419±0.0024 | **0.4388±0.0023** | **0.4136±0.0032** | **0.3891±0.0063** | **0.3653±0.0120** |
| | $\Delta_{\text{Err}}$ | CENET | 0.0697±0.0004 | **0.0647±0.0010** | **0.0596±0.0027** | 0.0621±0.0057 | 0.0678±0.0051 |
| | | WASSERSTEINNET | 0.0697±0.0004 | 0.0691±0.0006 | 0.0698±0.0011 | 0.0631±0.0022 | **0.0592±0.0033** |

**COMPAS**

| | | $\lambda$ | 0.0 | 0.1 | 0.5 | 1.0 | 5.0 |
|---|---|---|---|---|---|---|---|
| COMPAS | $R^2$ | CENET | 0.1631±0.0127 | 0.1610±0.0119 | 0.1542±0.0129 | **0.1515±0.0125** | 0.1418±0.0151 |
| | | WASSERSTEINNET | 0.1631±0.0127 | **0.1645±0.0136** | **0.1564±0.0125** | 0.1471±0.0151 | **0.1439±0.0143** |
| | $\Delta_{\text{Err}}$ | CENET | 0.0088±0.0048 | **0.0075±0.0044** | **0.0067±0.0046** | **0.0066±0.0039** | **0.0063±0.0046** |
| | | WASSERSTEINNET | 0.0088±0.0048 | 0.0088±0.0045 | 0.0079±0.0041 | 0.0070±0.0036 | 0.0072±0.0032 |

**Crime**

| | | $\lambda$ | 0.0 | 0.1 | 1.0 | 5.0 | 10 |
|---|---|---|---|---|---|---|---|
| Crime | $R^2$ | CENET | 0.5435±0.0077 | 0.5290±0.0107 | 0.1632±0.0573 | 0.1334±0.0720 | 0.1692±0.1509 |
| | | WASSERSTEINNET | 0.5435±0.0077 | **0.5467±0.0063** | **0.5472±0.0065** | **0.5446±0.0091** | **0.5319±0.0143** |
| | $\Delta_{\text{Err}}$ | CENET | 0.0191±0.0003 | **0.0175±0.0004** | 0.0230±0.0027 | 0.0221±0.0079 | 0.0265±0.0051 |
| | | WASSERSTEINNET | 0.0191±0.0003 | 0.0194±0.0004 | **0.0191±0.0004** | **0.0180±0.0005** | **0.0173±0.0010** |

**Law**

| | | $\lambda$ | 0.0 | 0.1 | 1.0 | 5.0 | 10 |
|---|---|---|---|---|---|---|---|
| Law | $R^2$ | CENET | 0.1197±0.0314 | **0.1200±0.0299** | **0.1059±0.0277** | **0.0464±0.0542** | **0.0235±0.0732** |
| | | WASSERSTEINNET | 0.1197±0.0314 | 0.1134±0.0339 | 0.0902±0.0292 | 0.0316±0.0476 | 0.0146±0.0553 |
| | $\Delta_{\text{Err}}$ | CENET | 0.0102±0.0010 | 0.0101±0.0009 | **0.0090±0.0018** | **0.0070±0.0030** | **0.0066±0.0030** |
| | | WASSERSTEINNET | 0.0102±0.0010 | **0.0098±0.0016** | 0.0090±0.0019 | 0.0072±0.0025 | 0.0069±0.0027 |

Table 3 shows $R^2$ regression scores and error gaps when $\lambda$ changes in CENET and WASSERSTEINNET. We see that the error gap gradually decreases with the increase of the trade-off parameter $\lambda$ in most scenarios with small accuracy loss (except for CENET in Adult dataset and Crime dataset when $\lambda$ is large), which demonstrates the overall effectiveness of our proposed algorithms. Plus, the increase of $\lambda$ generally leads to the instability of training processes with larger variances of both values of $R^2$ and error gap. In contrast to WASSERSTEINNET, CENET outperforms in mitigating the accuracy disparity while achieving similar or better accuracy in COMPAS and Law dataset. In Adult and Crime dataset, when $\lambda$ is small, CENET also does better in reducing the error gap than WASSERSTEINNET with similar accuracy loss. The results follow the fact that minimizing total variation distance between two continuous distributions ensures the minimization of Wasserstein distance (Gibbs & Su, 2002). However, when $\lambda$ increases, WASSERSTEINNET achieves better accuracy and performance disparity trade-off while CENET suffers significant accuracy loss and may fail to decrease the error gap. It is not surprising since the estimation of total variation in minimax optimization could lead to an unstable training process (Arjovsky & Bottou, 2017; Arjovsky et al., 2017).

Table 4 shows $R^2$ regression scores and error gaps when $\epsilon$ changes in BGL. We see that with the decrease of the trade-off parameter $\epsilon$, both the values of $R^2$ and error gaps decrease. This is because when upper bound $\epsilon$ of BGL is small, the accuracy disparity is also mitigated. When $\epsilon$ is above/below a certain threshold, the values of $R^2$ and error gaps then increase/decrease. It is also worth to note that the exponentiated-gradient approach to solve BGL does not introduce the randomness during optimization.

Table 4: $R^2$ regression scores and error gaps when $\epsilon$ changes in BGL.

| | $\epsilon$ | 0.07 | 0.1 | 0.2 | 0.5 |
|---|---|---|---|---|---|
| Adult | $R^2$ | 0.3508±0.0000 | 0.3508±0.0000 | 0.3696±0.0000 | 0.3696±0.0000 |
| | $\Delta_{\mathrm{Err}}$ | 0.0612±0.0000 | 0.0612±0.0000 | 0.0726±0.0000 | 0.0726±0.0000 |
| | $\epsilon$ | 0.1 | 0.2 | 0.3 | 0.5 |
| COMPAS | $R^2$ | 0.1478±0.0000 | 0.1478±0.0000 | 0.1507±0.0000 | 0.1507±0.0000 |
| | $\Delta_{\mathrm{Err}}$ | 0.0072±0.0000 | 0.0072±0.0000 | 0.0086±0.0000 | 0.0086±0.0000 |
| | $\epsilon$ | 0.01 | 0.02 | 0.03 | 0.05 |
| Crime | $R^2$ | 0.3922±0.0000 | 0.3922±0.0000 | 0.5380±0.0000 | 0.5380±0.0000 |
| | $\Delta_{\mathrm{Err}}$ | 0.0189±0.0000 | 0.0189±0.0000 | 0.0238±0.0000 | 0.0238±0.0000 |
| | $\epsilon$ | 0.01 | 0.02 | 0.03 | 0.05 |
| Law | $R^2$ | 0.1407±0.0000 | 0.1407±0.0000 | 0.1407±0.0000 | 0.1412±0.0000 |
| | $\Delta_{\mathrm{Err}}$ | 0.0094±0.0000 | 0.0094±0.0000 | 0.0094±0.0000 | 0.0101±0.0000 |

Table 5: $R^2$ regression scores and error gaps when $\epsilon$ changes in COD.

| | $\epsilon$ | 0.01 | 0.1 | 0.5 | 1.0 |
|---|---|---|---|---|---|
| COMPAS | $R^2$ | 0.1033±0.0111 | 0.1144±0.0100 | 0.1146±0.0099 | 0.1146±0.0099 |
| | $\Delta_{\mathrm{Err}}$ | 0.0064±0.0042 | 0.0083±0.0058 | 0.0085±0.0060 | 0.0085±0.0060 |
| | $\epsilon$ | 0.01 | 0.1 | 0.5 | 1.0 |
| Crime | $R^2$ | 0.1262±0.0000 | 0.3284±0.0000 | 0.3603±0.0000 | 0.3603±0.0000 |
| | $\Delta_{\mathrm{Err}}$ | 0.0312±0.0000 | 0.0307±0.0000 | 0.0343±0.0000 | 0.0343±0.0000 |
| | $\epsilon$ | 0.01 | 0.1 | 0.5 | 1.0 |
| Law | $R^2$ | 0.1262±0.0000 | 0.3284±0.0000 | 0.3606±0.0000 | 0.3603±0.0000 |
| | $\Delta_{\mathrm{Err}}$ | 0.0312±0.0000 | 0.0307±0.0000 | 0.0343±0.0000 | 0.0343±0.0000 |

Table 5 shows $R^2$ regression scores and error gaps when $\epsilon$ changes in COD. We see that with the decrease of the trade-off parameter $\epsilon$, both the values of $R^2$ and error gaps decrease. It is worth to note that the the optimization of QCQP to solve COD does not introduce the randomness, and the only randomness introduced in COMPAS dataset is because using the random Fourier features in prediction achieves the best performance in COMPAS dataset.

## C.2  VISUALIZATION OF TRAINING PROCESSES

We visualize the training processes of our proposed methods CENET and WASSERSTEINNET in the Adult dataset and COMPAS dataset in Figure 4 and Figure 5, respectively. We also compare their training dynamics with the model performance that we solely minimize the MSE loss (*i.e.*, $\lambda = 0$) and we term it as NO DEBIAS.

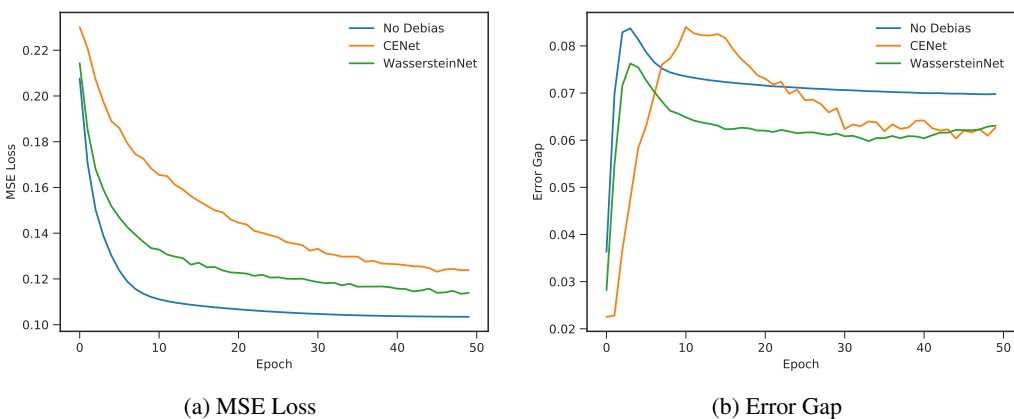

(a) MSE Loss                    (b) Error Gap

Figure 4: Training visualization of CENET, WASSERSTEINNET ($\lambda = 50$) and NO DEBIAS ($\lambda = 0$) in the Adult dataset.

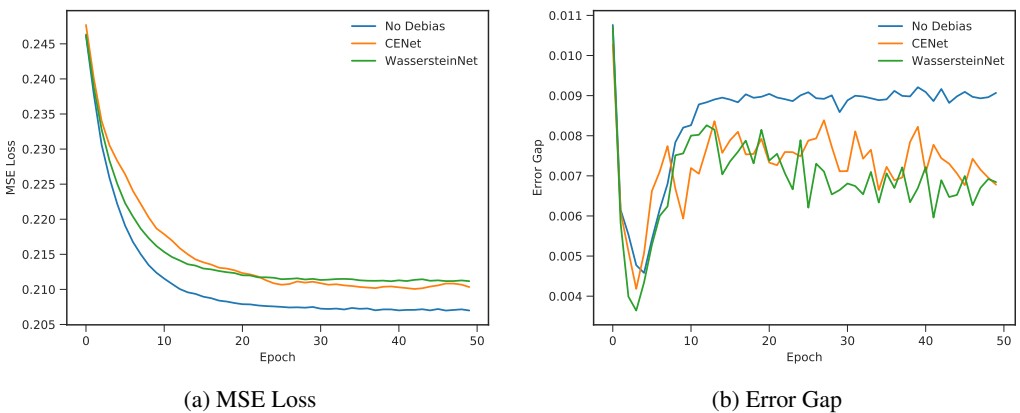

(a) MSE Loss

(b) Error Gap

Figure 5: Training visualization of CENET, WASSERSTEINNET ($\lambda = 5$) and NO DEBIAS ($\lambda = 0$) in the COMPAS dataset.

In Figure 4 and Figure 5, we can see that as the training progress, the MSE losses in both datasets are decreasing and finally converge. However, the training dynamics of error gaps are much more complex even in the NO DEBIAS case. Before convergence, the training dynamics of error gaps differs among different datasets. Our methods enforce the models to converge to the points where error gap are smaller while preserving the models' predictive performance. It is also worth to note that minimax optimization makes the training processes somehow unstable.

