# OpenReview forum: "Understanding and Mitigating Accuracy Disparity in Regression"
_ICLR.cc/2021/Conference — Reject_

### Official Review · AnonReviewer4 · 2020-10-27
**Weak theoretical results and suspicious experimental results**

**Rating:** 4
**Confidence:** 3

**Review:**

This paper deals with a fair regression problem in which the accuracy disparity is employed as a fairness measure. The authors derived the upper and lower bounds on the difference of accuracy between groups to demonstrate that imbalance in the groups' sizes leads to accuracy disparity. Furthermore, they propose learning algorithms enabling us to mitigate the accuracy disparity, which is accomplished by minimizing the upper bound they derived. The empirical evaluations show that the present methods achieve a better trade-off between accuracy and fairness than some existing fair regression methods.


The strong points are as follows:
- The authors tackle an important and interesting problem, disparate mistreatment in the regression problem, which is not much investigated so far.
- The authors give some theoretical insight into the hypothesis that the imbalanced groups lead to unfairness by ignoring the minor group.

The weak points are as follows:
- The theoretical results are somewhat weak to support the claim that the imbalance groups lead to the accuracy disparity.
- This paper lacks a comparison with some important existing methods that share the same concept for mitigating unfairness.
- There is something suspicious in the experimental results.

Overall, my recommendation is rejection because the theoretical and experimental results are somewhat weak to support the authors' claims. Also, I have concerns that the experimental results are something wrong.

The theoretical contribution the authors claim to make is to clarify the cause of the accuracy disparity. The cause here is the imbalance in groups, as discussed in the geometric interpretation paragraph on page 4.  Theorem 3.1 and Theorem 3.2 give insight into the authors' claim; however, they are slightly weak to support the statement that the group imbalance yields accuracy disparity. Because Theorem 3.1 and Theorem 3.2 only provide the upper and lower bounds, the feasible region can be an arbitrary shape in the green area in Fig. 1. The possibility remains that the actual feasible region is cone-like shape such that its apex is on the diagonal line. In this case, the minimization of the overall error can result in matched accuracy. Indeed, there is a counterexample of the claim. Suppose the hypothesis class consists of functions that output a constant, and the conditional variances $\mathrm{Var}[Y|A]$ are equivalent for any $A$. Then, the optimal regressor, which minimizes the overall mean squared error, achieves the matched accuracy because $\mathrm{Err}_{\mathcal{D}_a}= \mathrm{Var}[Y|A=a]$ in this case. I guess some assumptions on the hypothesis class and underlying distribution are necessary to prove the authors' claim.

The authors design the proposed algorithm to match the distribution over the representation conditioned on the true outcome and sensitive attribute. This requirement is equivalent to the equalized odds. There are several works for tackling the fair regression under the equalized odds constraint, including
- J. Mary et al. Fairness-Aware Learning for Continuous Attributes and Treatments. In ICML'19.
- H. Narasimhan et al. Pairwise Fairness for Ranking and Regression. In AAAI'20.
Since these methods' design concept is equivalent to the presented one, the authors should clarify their method's merits compared with these existing methods.

I'm very suspicious about the experimental results of BGL and CoD. We can find in the original papers of BGL and CoD that these methods can achieve $R^2$ from 0.32 to 0.64 in the crime dataset by choice of their hyperparameters. These values are calculated by my hand from the MSE shown in these papers using the relationship $R^2 = 1 - \mathrm{MSE} / \mathrm{Var}[Y]$. However, in the experimental result shown in Fig. 2 (c), the values of $R^2$ for BGL and CoD are at most 0.55.

In Section 4.2, the authors claim that Fig. 2 shows that trade-offs exist between accuracy and fairness. However, I cannot find such an inclination from Fig. 2.

It is unclear that the presented methods can achieve a better fairness-accuracy trade-off than the existing fair representation methods. The fair representation methods may work in the regression setting with small modifications, even if the original ones are designed for the classification setting. Hence, these methods can apply to this paper's setting.

### Minor comments
- Why are the Game-Theoretic Interpretation helpful? Why can we obtain new insights by interpreting the Eqs. (1) and (3) in a game-theoretic manner? I cannot find any new insights from the game-theoretic interpretation.

---

> ### Author Response · Authors · 2020-11-16
> **Response to Reviewer #4, Part 1**
>
> 1. Justification of our theoretical results.
>
> Thanks for your insightful comments on our geometric interpretation. The provided counter example could be covered by Theorem 3.3: When the hypothesis outputs a constant (the second term in the upper bound in Theorem 3.3 then equals to zero) and $\text{Var}[Y|A]$ is equivalent for any $A$ (the first term in the upper bound in Theorem 3.3 then equals to zero), then the upper bound in Theorem 3.3 equals to zero. In this case the green area will shrink to a ray and the optimal regressor which minimizes the overall mean squared error can achieve accuracy parity.
>
> Note that we provide two upper bounds for accuracy disparity in Theorem 3.2 and Theorem 3.3 and they complement each other: Theorem 3.2 decomposes accuracy disparity into the distance of input distribution across groups, noise (variance) difference and the discrepancy of optimal decision functions across different groups, while Theorem 3.3 decomposes accuracy disparity into the distance between label distributions and the discrepancy between conditional predicted distributions across groups.
>
>
> 2. Comparison with other works
>
> 2a. J. Mary et al. Fairness-Aware Learning for Continuous Attributes and Treatments. In ICML'19: The paper proposed to use the Hirschfeld-Gebelein-R´enyi Maximum Correlation Coefficient (HGR) to generalize fairness measurement to continuous variables. To be specific, they ensured equalized odds (demographic parity) constraint by minimizing the $\chi^2$ divergence (the relaxation of HGR) between the predicted variable and the sensitive variable (conditioned on target variable).
>
> As a comparison, our theoretical analysis quantifies the difference of different (conditional) distributions in total variation distance and Wasserstein distance.  According to (Gibbs, A. L., & Su, F. E. 2002), under mild assumptions about the underlying density functions, the convergence of TV distance as well as the Wasserstein distance leads to a tighter analysis than the one based on $\chi^2$ divergence. In addition, they proposed to estimate the $\chi^2$ divergence based on Witsenhausen’s characterization mixed with a Kernel Density Estimation (KDE) while we estimate the TV distance and Wasserstein distance using adversarial representation training techniques.
>
> 2b. H. Narasimhan et al. Pairwise Fairness for Ranking and Regression. In AAAI'20:
> The paper proposed pairwise fairness notions (e.g., pairwise equal opportunity requires each pair from two arbitrary different groups to be equally-likely to be ranked correctly) for ranking and regression models. They formulate the training with fairness goals as a constrained optimization problem and solve the optimization under the proxy-Lagrangian framework.
>
> Compared to their work, our fairness notions and methods are different from theirs: we propose to align (conditional) distributions across groups to reduce accuracy disparity using minimax optimization.
>
>
> 3. Justification of the experimental results.
>
> 3a. Different CoD and BGL results from the original papers.
>
> The CoD and BGL results are different from the original paper because our experiment setups (e.g., data preprocessing pipelines) are different from theirs. In the CoD paper, they use two numerical sensitive attributes (the ratio of African American people and foreign-born people, respectively) in their experiment; In the BGL paper, they use race (whether the majority population of the community is white) as the sensitive attribute. In addition, the BGL paper does not provide the details of how they preprocess the crime dataset (e.g., whether they have removed the sensitive attribute from the input, the proportion of the training and test set). As a comparison, we remove all the sensitive attributes related to race from the model input in the crime dataset and set our binary sensitive attribute as 1 if the population percentage of the white is greater or equal to 80% otherwise 0.
>
> So the numbers reported in their papers and ours are not directly comparable due to the different experimental setups. For more details of our experiment setups, please refer to Appendix B. For the implementation of our baselines and algorithms and the replication of our results, we have provided our codes in the supplementary materials.
>
> 3b. In Section 4.2, the authors claim that Fig. 2 shows that trade-offs exist between accuracy and fairness. However, I cannot find such an inclination from Fig. 2.
>
> In figure 2, the general trend is with the decrease of the values of error gaps, the values of $R^2$ also decrease for each method. The trend is clear in COMPAS (Figure 2(b)) and law dataset (Figure 2(d)). In the Adult dataset (Figure 2(a)) and crime dataset (Figure 2(c)), except for the CENet (blue points), all other methods follow the trend. The reason why CENet fails the trend is due to the optimization problem (Arjovsky, M., & Bottou, L. 2017) (See Appendix C for more detailed discussion on this).

---

> > ### Author Response · Authors · 2020-11-16
> > **Response to Reviewer #4, Part 2**
> >
> > 3c. It is unclear that the presented methods can achieve a better fairness-accuracy trade-off than the existing fair representation methods. The fair representation methods may work in the regression setting with small modifications, even if the original ones are designed for the classification setting. Hence, these methods can apply to this paper's setting.
> >
> > To the best of our knowledge, most fair representation methods aim to focus on demographic parity or equalized odds in the classification. One work (Zhao et al., 2019) tries to address the accuracy disparity problem in classification via representation methods: they propose to align conditional distribution and minimize balanced error rate to mitigate accuracy disparity, where they need one discriminator for each value of $Y$. However, it is infeasible to do so directly in regression: since the $Y$ can take on infinite values in regression, neither conditional distribution could be aligned nor balanced error rate could be minimized directly.
> >
> > 4. “Why are the Game-Theoretic Interpretations helpful? Why can we obtain new insights by interpreting the Eqs. (1) and (3) in a game-theoretic manner? I cannot find any new insights from the game-theoretic interpretation.”
> >
> > The Game-Theoretic Interpretations (Figure 1(b)) illustrate the idea of our algorithms and it helps the reader outside our area to better understand our algorithm. They also  highlight the key of our algorithm is to add external label information to align (conditional) distribution across groups to mitigate accuracy disparity.
> >
> >
> > Gibbs, A. L., & Su, F. E. (2002). On choosing and bounding probability metrics. International statistical review, 70(3), 419-435.
> >
> > Arjovsky, M., & Bottou, L. (2017). Towards principled methods for training generative adversarial networks. arXiv preprint arXiv:1701.04862.
> >
> > Zhao, H., Coston, A., Adel, T., & Gordon, G. J. (2019, September). Conditional Learning of Fair Representations. In International Conference on Learning Representations.

---

### Official Review · AnonReviewer3 · 2020-10-29
**Good result and writing but better differentian to fair representation in classification is needed.**

**Rating:** 6
**Confidence:** 3

**Review:**

This paper has two parts: (i) in the first part the authors provide an upper and a lower bound for error disparity among groups in the regression setup. (ii) In the second part, they propose an adversarial learning method to mitigate the disparate error among groups. Mainly they focus on reducing the error disparity by reducing the distance between the joint distribution of Z (the new learned representation) and Y between the two groups. The paper was finely written, and it was interesting. I will address my concerns regarding this paper as follows:

First part:

I did not understand the importance of Theorem 3.1 the lower bound is dependent on the classifier (thus is not inherent) and the authors only focused on the first term and did not talk about the second term.

Related work: There is some related work after Chen 2018 paper which focuses on understanding the source of error disparity among groups (e.g., https://proceedings.icml.cc/static/paper_files/icml/2020/1320-Paper.pdf which also points out the difference between the distribution of the groups or https://arxiv.org/pdf/1906.08386.pdf ). I think the authors should explain the other work on understanding the source of error disparity.


Second part:

My main concern regarding this part is that I feel like the authors did not place their work accurately among the related work for the fair representation part. In particular, why we cannot readily use Madres 2018 approach for the regression task. I think the authors should explain why the regression setup provides new challenges on fair learning in comparison to the classification setup. In the experiment section, for the binary datasets do we get the same result as this paper if we use the fair representation methods in classification? What is inherently different here.


Minor suggestion: I got a bit confused about equation 3, you wrote that the signature of f is Z-> R but it seems that you are trying to minimize joint distribution. I think instead of the game-theoretic interpretation (which is kind of clear) if you can expand on this and why the optimization is easier is better.

---

> ### Author Response · Authors · 2020-11-16
> **Response to Reviewer #3**
>
> 1. Question on Theorem 3.1 (“I did not understand the importance of Theorem 3.1 the lower bound is dependent on the classifier (thus is not inherent) and the authors only focused on the first term and did not talk about the second term.”)
>
> The second term in Theorem 3.1 indicates the discrepancy between induced distribution of two groups by the predictor: if the predictor satisfies demographic parity, the second term will equal to zero. The lower bound means if the predictor satisfies demographic parity but the difference between the label distribution across groups is large, then the lower bound will get larger. In a nutshell, a predictor which only satisfies demographic parity does not lead to accuracy parity if the discrepancy between the label distribution in different groups is large.
>
>
> 2. Comparison with related works:
>
> 2a. Feature Noise Induces Loss Discrepancy Across Groups, ICML 2020: The paper analyzed that the loss difference among different groups is determined by the amount of latent (unobservable) feature noise and the difference between means, variances, and sizes of the groups with an assumption that there are a latent random feature and a noise feature that play roles in the generation of the observable features.
>
> As comparison, we only assume there is a joint distribution $\mathcal{D}$ over $X$, $Y$ and  $A$ from which the data are sampled. Through the lens of information theory, we propose an error decomposition theorem that decomposes accuracy disparity into the distance between label populations and the distance between conditional representations.
>
>
> 2b. Inherent Tradeoffs in Learning Fair Representations, NeurIPS 2019: The paper proposed an error decomposition theorem (Theorem 3.3 in their paper) for accuracy disparity in binary classification. However, their error decomposition theorem does not lead to algorithm interventions to mitigate accuracy disparity in classification settings.
>
> In contrast, we propose an error decomposition theorem for accuracy disparity in regression and it suggests minimizing the discrepancy of the conditional predicted distribution across groups to mitigate the problem. Moreover, we provide two practical algorithms to mitigate the problem.
>
> 3. Comparison with Madres 2018 approach and the challenges of mitigating accuracy disparity in regression using representation learning (“Why we cannot readily use Madres 2018 approach for the regression task. I think the authors should explain why the regression setup provides new challenges on fair learning in comparison to the classification setup. In the experiment section, for the binary datasets do we get the same result as this paper if we use the fair representation methods in classification? What is inherently different here”).
>
> Madres et al., 2018 proposed a general framework to learn a fair and transferable representation for demographic parity and equalized odds in binary classification instead of accuracy disparity (see figure 1 in their paper). In contrast, we propose to align (conditional) distributions across groups to reduce accuracy disparity in regression.
>
> In fact, prior to this work, it is not clear aligning conditional distributions could lead to (approximate) accuracy parity. Zhao et al., (2019) aimed to address the accuracy disparity problem in classification via adversarial representation methods and they proposed to align conditional distributions and minimize balanced error rate to mitigate accuracy disparity. In their algorithm design, they need one discriminator for each value of $Y$ to perform the estimations. However, it is infeasible to do so directly in regression: since the $Y$ can take on infinite values in regression. Thus, aligning conditional distributions or minimizing balanced error rate could be achieved in the regression setting directly.
>
> 4. Typo problem (“I got a bit confused about equation 3, you wrote that the signature of f is Z-> R but it seems that you are trying to minimize joint distribution.”).
>
> It is a typo here: it should be $Z \times Y \to R$.

---

### Official Review · AnonReviewer1 · 2020-10-29
**This work proposes an adversarial training procedure for reducing disparity in fairness of regression tasks. The main contributions are theoretical bounding the disparity using distance between label populations and conditional distributions. The bounds are used to motivate the adversarial training cost function. Overall the paper is well written and motivated. Experimental results are reasonable. I have a few comments in the following for clarification below.**

**Rating:** 7
**Confidence:** 3

**Review:**

1. Figure 1 (left) is illustrative and insightful. I suggest authors include a simulated/semi-synthetic dataset (semi-synthetic versions can be created using the datasets that are already being used for experiments) and show clearly that their adversarial training procedure is likely to learn hypothesis within the feasible region that guarantees or is close to minimizing the disparity. I think the cost function and training procedure will have additional grounding with such analysis.

2. Missing citation - https://arxiv.org/abs/1901.10566 and comparison in related work.

3. "Nevertheless, throughout this paper we mainly focus accuracy parity as our fairness notion, due to the fact that three widely used commercial face recognition systems have been shown to exhibit substantial accuracy disparities between different demographic subgroups (Buolamwini & Gebru, 2018). This observation has already brought huge public attention (e.g., see New York Times, The Verge, and Insurance Journal) and calls for commercial face recognition systems that (at least approximately) satisfy accuracy parity". I have concerns about this claim in the paper. The takeaway of highlighted problems with commercial face recognition is not that that the methods will be ready for practical use by ensuring approximate accuracy parity. I suggest the authors remove this motivation as facial recognition as a system is fraught with many other societal challenges that cannot be resolved by ensuring parity of performance.

4. Please be more elaborate in proofs in the appendix rather than having the reader fill in many gaps.

5. What is the sensitivity of regularization to performance and how was the parameter selected?

6. Minor -
typos in appendix:
1.  "follows immediately the triangle inequality and Lemma" in proof of Thm 3.1
2.  "Now it is suffice to bound the term" proof of Thm 3.3

---

> ### Author Response · Authors · 2020-11-16
> **Response to Reviewer #1**
>
> 1. Visualization of training process (I suggest authors include a simulated/semi-synthetic dataset (semi-synthetic versions can be created using the datasets that are already being used for experiments) and show clearly that their adversarial training procedure is likely to learn hypothesis within the feasible region that guarantees or is close to minimizing the disparity. I think the cost function and training procedure will have additional grounding with such analysis.)
>
> Thanks for your suggestions. We have visualized adversarial training procedures and put the corresponding results and analysis in the appendix C in the updated version of our paper.
>
> 2. Comparison with other works
>
> 2a. Fair Regression for Health Care Spending: The paper considered regression problems in health care spending and proposed five fairness criteria (e.g., covariance constraint, net compensation penalization) in the healthcare domain. They formulated the problems as constrained optimization problems and use off-the-shelf disciplined convex programming tools to solve them.
>
> Compared to their work, we focus on reducing regression disparity in regression through (conditional) distribution alignment using adversarial training techniques.
>
> 3. The takeaway of highlighted problems with commercial face recognition is not that the methods will be ready for practical use by ensuring approximate accuracy parity. I suggest the authors remove this motivation as facial recognition as a system is fraught with many other societal challenges that cannot be resolved by ensuring parity of performance.
>
> Thanks for your suggestions. We have deleted this example in our updated version of paper.
>
> 4. Please be more elaborate in proofs in the appendix rather than having the reader fill in many gaps.
>
> Thanks for your suggestions. We have added more elaborations in the proofs in our updated version of paper.
>
> 5. What is the sensitivity of regularization to performance and how was the parameter selected?
>
> In the Table 1 in the appendix, we can see that the sensitivity of regularization differs among different datasets: The model performance in the Adult dataset is least sensitive to the change of regularization while other datasets are more sensitive.
>
> The selection of the value of the regularization parameter depends on the actual use cases: if we would like to reduce accuracy disparity to a large extent, we will increase the value of the regularization parameter; otherwise we just need small values of the regularization parameter.
>
> 6. Typos: Minor - typos in appendix.
>
> Thanks for pointing out the typos. We have corrected them in the updated version.

---

### Official Review · AnonReviewer2 · 2020-10-29
**Overall, I vote for accepting. The discovery between accuracy parity and the distribution gaps across groups is interesting.**

**Rating:** 6
**Confidence:** 3

**Review:**

This paper theoretically and empirically studies accuracy disparity in regression problems. It proves an information-theoretic lower bound on the joint error and a complementary upper bound on the error gap across groups to depict the feasible region of group-wise errors. It further proposes to achieve accuracy parity theoretically and empirically by learning conditional group-invariant representations using statistical distances.

Overall, I vote for accepting. The discovery between accuracy parity and the distribution gaps across groups is interesting.

Pros:
    1. The paper provides a deeper understanding of the accuracy parity, which is interesting to me and motivates the proposed algorithms.

Cons:
    1. The motivation for the studied problem is not very clear to me. I know the importance of both regression and accuracy parity. But why we want to consider the combination of these two notions? What is the main difference between considering classification with accuracy parity?
    2. Page 4, Geometric Interpretation. Mention Theorem 3.3 by a mistake?

---

> ### Author Response · Authors · 2020-11-16
> **Response To Reviewer #2**
>
>
> 1. Motivation of our work (“why do we want to consider the combination of these two notions? What is the main difference between considering classification with accuracy parity?”)
>
> Compared to the accuracy disparity problem in classification settings, accuracy disparity in regression is a less studied but also important problem. For example, in a healthcare spending prediction system which predicts the estimated healthcare payments to insurance plans, we do not want the prediction error gap to be too large among different demographic subgroups.
>
> 2. Typo problem. (“In Page 4, Geometric Interpretation. Mention Theorem 3.3 by a mistake?”)
>
> Yes, this is a typo, it should be Theorem 3.2 instead.

---

### Author Response · Authors · 2020-11-16
**General Response**

We thank all the reviewers for the thoughtful comments and we answer each reviewer’s questions individually below. In summary, we have the following major updates in our rebuttal:

1. We made detailed comparisons with the related works in our rebuttal and cited the papers in our updated pdf draft(Reviewers #1, #3, #4);
2. We justified that our experiment baseline results are different from the original paper due to the differences in experimental setups and uploaded our codes for the replication purpose (Reviewer #4);
3. We justified our theoretical results by covering the counter example using Theorem 3.3 and explained how the feasible region of our geometric interpretation looks like if the upper bound of the error gap shrinks to zero (Reviewer #4);
4. We presented the challenges of mitigating accuracy disparity using fair representation approaches in regression and explained why those fair representation approaches using in fair classification cannot be directly applied to regression (Reviewer #3, #4);
5. We fixed the typos (Reviewer #1, #2, #3).

---

### Decision · Program_Chairs · 2021-01-07
**Final Decision**

**Decision:**

Reject

**Comment:**

This papers considers the problem of accuracy disparity in regression for the case of binary sensitive attributes. It provides bounds for accuracy disparity and introduces two methods to enforce this criterion based on representation learning.

The reviews are in agreement that the paper is generally clear and well written, but have different opinions regarding the significance of the contribution and the experimental section. I did read the paper with care myself and overall I do share the concerns raised by Reviewers 3 and 4 that the paper does not place itself accurately wrt to the current literature, both in the discussion and the experimental section. The response to the reviewers about methods that can achieve accuracy disparity for classification is not satisfactory, also considered that two of the analysed datasets are about binary classification tasks. Regarding the results on these datasets, it would be useful to report classification accuracy rather than (in addition to) R^2.
The proposed methods do not seem to show a significant advantage versus the methods considered for comparison.

Minor comments:
The proposed methods are inspired by Theorem 3.2. However, is not enforcing accuracy disparity by minimising some distance between conditional distributions what the literature does? I cannot think of other meaningful ways to achieve this criterion. In fact, referring to this theorem, the authors says 'However, it is nearly impossible to collect noiseless data with group-invariant input distribution. Moreover, there is no guarantee that the upper bound will be lower if we learn the group-invariant representation that minimizes dTV(D0(X), D1(X)) alone, since the learned representation could potentially increase the variance. In this regard, we prove a novel upper bound which is free from the above noise term to motivate aligning conditional distributions to mitigate the error disparity'. But minimizing dTV(D0(X), D1(X)) might not be desirable if the dependence between the sensitive attribute A and the data is considered legitimate. The point of using conditional distributions is to allow that dependence to be retained.